# A logical network-based drug-screening platform for Alzheimer's disease representing pathological features of human brain organoids

Jong-Chan Park[1,2,3,4,11], So-Yeong Jang [5,11], Dongjoon Lee[1,3,11], Jeongha Lee[1], Uiryong Kang [5], Hongjun Chang[5], Haeng Jun Kim[1,3], Sun-Ho Han[1,2,3], Jinsoo Seo [6], Murim Choi [1], Dong Young Lee[7,8,9], Min Soo Byun[10], Dahyun Yi [7], Kwang-Hyun Cho [5✉] & Inhee Mook-Jung [1,2,3✉]

Developing effective drugs for Alzheimer's disease (AD), the most common cause of dementia, has been difficult because of complicated pathogenesis. Here, we report an efficient, network-based drug-screening platform developed by integrating mathematical modeling and the pathological features of AD with human iPSC-derived cerebral organoids (iCOs), including CRISPR-Cas9-edited isogenic lines. We use 1300 organoids from 11 participants to build a high-content screening (HCS) system and test blood–brain barrier-permeable FDA-approved drugs. Our study provides a strategy for precision medicine through the convergence of mathematical modeling and a miniature pathological brain model using iCOs.

---

[1] Department of Biochemistry and Biomedical Sciences, College of Medicine, Seoul National University, Seoul 03080, Republic of Korea. [2] Neuroscience Research Institute, Medical Research Center, College of Medicine, Seoul National University, Seoul 03080, Republic of Korea. [3] SNU Dementia Research Center, College of Medicine, Seoul National University, Seoul 03080, Republic of Korea. [4] Department of Neurodegenerative Disease, UCL Queen Square Institute of Neurology, University College London, London WC1N 3BG, United Kingdom. [5] Department of Bio and Brain Engineering, Korea Advanced Institute of Science and Technology (KAIST), Daejeon 34141, Republic of Korea. [6] Department of Brain and Cognitive Science, Daegu Gyeongbuk Institute of Sciences and Technology (DGIST), Daegu 42988, Republic of Korea. [7] Institute of Human Behavioral Medicine, Medical Research Center, Seoul National University, Seoul 03080, Republic of Korea. [8] Department of Psychiatry, College of medicine, Seoul National University, Seoul 03080, Republic of Korea. [9] Department of Neuropsychiatry, Seoul National University Hospital, Seoul 03080, Republic of Korea. [10] Department of Neuropsychiatry, Seoul National University Bundang Hospital, Seongnam 13620, Republic of Korea. [11] These authors contributed equally: Jong-Chan Park, So-Yeong Jang, Dongjoon Lee. ✉email: ckh@kaist.ac.kr; inhee@snu.ac.kr

Alzheimer's disease (AD) is the most common form of dementia, afflicting more than 40 million individuals worldwide[1]. Symptoms include memory deterioration and cognitive impairment, resulting from neuronal loss, hippocampal atrophy, and brain inflammation[2]. On the molecular level, the pathology is characterized mainly by the accumulation of amyloid plaques, neurofibrillary tangles, and dystrophic neurites consisting of hyper-phosphorylated tau protein[3], which can be identified by Pittsburgh compound B (PiB)-positron emission tomography (PET) and tau-PET, respectively[4]. The difficulty of acquiring human brain samples and the lack of a disease model that adequately recapitulates the pathological hallmarks pose significant challenges in the field.

Numerous animal and cellular models have been developed to tackle this issue, but each has unresolved limitations. Animal models provide valuable insights into disease mechanisms, but the commonly used transgenic rodent models carry familial AD mutations that account for 5% of all AD cases, although the most common type of AD is sporadic cases (>95%) in which apolipoprotein E (ApoE) ε4 allele is the major genetic risk factor for sporadic AD[5]. Moreover, AD phenotypes do not appear spontaneously with aging in non-transgenic mice, casting doubt on the existence of disease-initiating molecular pathways in these species[6]. Cellular models, such as iPSC-derived neurons, offer the advantage over animal models in having a human genetic background[7,8], but conventional monolayer culture, which lacks the interstitial space, fails to show the extracellular amyloid aggregate deposition that is a major hallmark of AD[9,10]. These limitations could explain the consistent clinical failure of medications found to be effective in pre-clinical models[11,12]. A physiologically relevant human-derived in vitro model for drug screening is urgently needed to enable the successful translation of AD drug candidates from bench to bedside.

In recent years, the development of cerebral organoids has opened up a previously unknown realm of the human brain that could not be explored due to limited accessibility[13,14]. Several groups have demonstrated the applicability of cerebral organoids in neurodegenerative studies by integrating genetic mutations that are definite causes of the disease[10,15]. These studies have shown that using cerebral organoids as a three-dimensional (3D) in vitro model can provide aspects of and insights into pathological conditions that could not be recapitulated in conventional monolayer culture. These include extracellular deposition of amyloidogenic peptides, propagation of protein through complex cell-to-cell interactions in a spatial context, and the impaired interplay of diverse cellular subtypes[16]. The guided formation of cerebral organoids by the timed supplementation of cells with defined growth factors has been shown to yield multiple neuronal subtypes at consistent proportions[14,17]. Using this method, we were able to produce a massive number of homogeneous organoids suitable for high-content screening (HCS) system.

AD is a multifactorial disease that is caused by malfunctions in its complex regulatory processes, such as vesicle trafficking, endocytosis, lipid metabolism, and immunity[18–20]. Several subtypes of AD exist according to their different onset mechanisms that depend on causal risk factors, which signifies the need to identify optimal drug target for each risk factor[21,22].

Given the complexity arising from the diverse risk factors and multi-step pathogenic processes of AD, it is difficult to identify a disease-modifying target for each patient by conventional single pathway analysis. Hence, an integrated system-level approach is required to determine an optimal drug target with the consideration of the existing genetic factors and their effects on the complex molecular landscape[23].

In this study, we show that iPSC-derived cerebral organoids (iCOs) developed from sporadic AD patients who are predisposed for an increased brain burden of both amyloid and tau, recapitulate the pathological features of the disease. Mass production of iCOs that are uniform in size and homogeneous in cell composition enabled us to perform drug screening using HCS system on a physiologically relevant platform. Mathematical modeling considering a network of molecular pathways and relevant genetic factors was employed to identify several FDA-approved drugs as candidates for drug repositioning. In sum, by integrating mathematical modeling and pathological features of brain organoids, we herein developed a drug-screening platform that can be expanded for use in precision medicine.

## Results

**The overall scheme of this study and demographics of participants**. The overall scheme of this study is presented in Fig. 1. Briefly, to mimic the brain of sporadic AD patients, iCOs were generated from PiB-negative or PiB-positive participants (PiB iCOs: PiB⁻ iCOs and PiB⁺ iCOs). iCOs from CRISPR-Cas9-edited apolipoprotein E (ApoE) ε4 isogenic iPSC lines were also used (ApoE iCOs: E3^par parental iCOs and E4^iso isogenic iCOs). The process of making sAD organoids is summarized in Fig. 2a as a flow-chart. The demographic information of participants is shown in Supplementary Table 1. After checking the pathological features of iCOs, we performed network modeling and perturbation analysis, and used the results to select candidate drugs. We tested the efficacy of the candidate drugs by HCS system and neuronal cell death assays. All experiments and analyses are detailed in Figs. 2–6.

**Characterization of the generated iPSCs and iCOs**. The methods used to generate the iPSCs and iCOs are presented in the Methods. The generated iPSCs showed alkaline phosphatase (ALP) positivity (Supplementary Fig. 1a) and expressed stem cell marker proteins, such as Oct4, Tra1-60, and Sox2 (Supplementary Fig. 1b, c). We also checked the karyotypes (Supplementary Fig. 1d). All of the generated iPSCs had normal karyotypes except for PiB⁺ #1 iPSCs, which showed triple X syndrome, and were thus excluded from the comparative analyses of PiB⁻ iCOs vs PiB⁺ iCOs. We also quality-checked the generated iCOs. On day 60, the iCOs showed some regions of Sox2⁺ neuroepithelial budding zones and had NeuN⁺ or beta III-tubulin⁺ neuronal cells (Supplementary Fig. 2a). We also performed RNA sequencing using both PiB iCOs and ApoE iCOs. Their total mRNA expression levels were stable, and no difference was observed between the iCOs (Supplementary Fig. 2b–e). PiB iCOs showed 1418 up-regulated and 247 down-regulated differentially expressed genes (DEG) patterns, and ApoE iCOs showed 1573 up-regulated and 545 down-regulated DEG patterns (Supplementary Fig. 2f, g). The top 10 diseases related to the DEGs, as determined by Gene Ontology (GO) analysis, included 'AD dementia' for both PiB iCOs and ApoE iCOs (Supplementary Fig. 2h). These results verified that our iPSCs and iCOs were well qualified and ready for our experiments.

**Pathological features of sAD iCOs**. To confirm that the iCOs could show pathological phenotypes, we checked the levels of pathogenic proteins, such as beta-amyloid (Aβ)1-42, Aβ1-40, total tau, and phosphorylated tau (p-tau), secreted to conditioned media. As expected, PiB⁺ iCOs secreted higher levels of these proteins than PiB⁻ iCOs (Fig. 2c, upper), and all of these levels were significantly correlated to the corresponding degree of real brain Aβ deposition, which was represented by the Pittsburgh compound B-positron emission tomography (PiB-PET) standardized uptake value ratio (SUVR) (Fig. 2d). Interestingly, among the PiB⁺ iCOs, ApoE ε4 carriers secreted more pathogenic

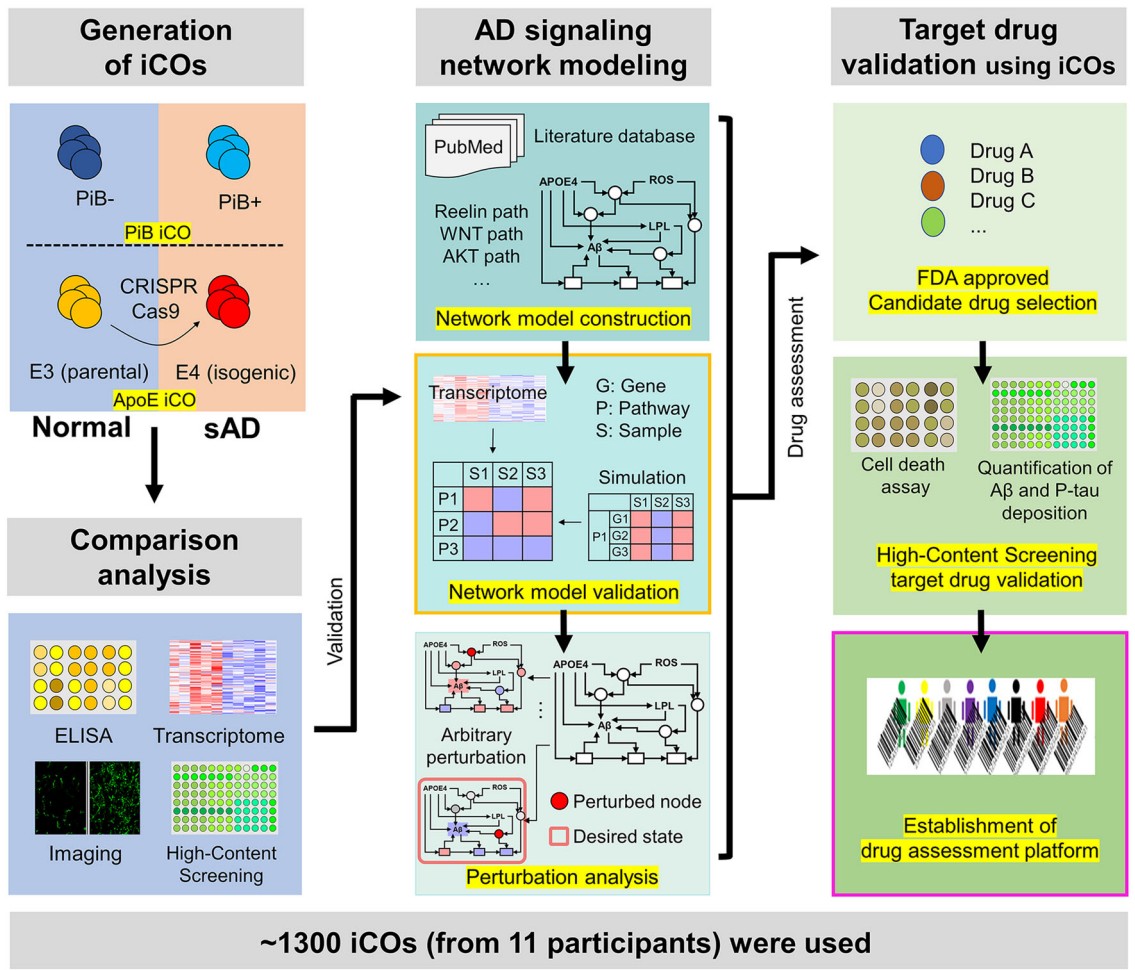

**Fig. 1 The overall scheme of this study.** To establish drug assessment platform for Alzheimer's disease (AD), three steps were performed: (i) Generation of iPSC-derived organoids (iCOs) both from normal and sporadic AD (sAD) participants, and CRISPR-Cas9 ApoE4 isogenic lines were used. (ii) Systems biology-based AD pathway simulation; Signaling network construction, network model validation, and identification of control nodes steps were conducted. (iii) Validation of drugs from the simulation; FDA-approved drugs were used, and the degree of AD pathogenesis was quantified by the high-contents screening (HCS) imaging system. Approximately, 1300 organoids from 11 participants were used for the drug assessment platform. PiB Pittsburgh compound B, ApoE apolipoprotein E, ELISA enzyme-linked immunosorbent assay, Aβ beta-amyloid, p-tau phosphorylated tau.

proteins, compared to ApoE ε4 non-carriers (Fig. 2c, lower). To examine this in greater depth, we generated iCOs from E3 parental (E3$^{par}$) and E4 isogenic (E4$^{iso}$) iPSCs, and compared them. Similar to the above-described results, E4$^{iso}$ iCOs exhibited higher levels of the examined pathogenic proteins, compared to E3$^{par}$ iCOs (Fig. 2e), with the exception of the total tau levels. We also tested whether our iCOs had neural activities. Our calcium oscillation assay revealed that the iCOs showed intracellular changes of calcium signaling. As previous reports found that sAD patients could have abnormal increases in intracellular calcium with neuronal hyperactivation[24], PiB$^+$ iCOs and E4$^{iso}$ iCOs showed higher calcium fluorescence and more calcium peaks than PiB$^-$ iCOs and E3$^{par}$ iCOs (Fig. 2f, g). This suggested that PiB$^+$ and E4$^{iso}$ iCOs might have abnormal calcium regulation. Next, we performed RNA sequencing and compared their overall mRNA expression patterns. Interestingly, our principal component analysis (PCA) plot showed that there was ApoE ε type-dependent separation (ApoE ε4 non-carriers, green; ApoE ε4 carriers, yellow) in the mRNA expression pattern (Fig. 2h). An exception was the PiB$^+$ #3 iCO, which was an ApoE ε4 non-carrier but carried a single-nucleotide polymorphism (SNP) variant (A288T, G > A) in lipoprotein lipase (LPL) (Supplementary Fig. 3a, b). It held an ApoE ε

type-independent position, but remained well separated from the PiB$^-$ iCOs. Consistent with a previous report[25], PiB$^+$ and E4$^{iso}$ iCOs showed numerous down-regulated DEGs related to synaptic functions and neurogenesis (Fig. 2i). Furthermore, our transcriptome data was further verified by comparing with public transcriptome data from the GEO database (Supplementary Fig. 4). We found a public iPSC-derived neuron data (PIN) for AD (Accession number: GSE143951, Platform number: GPL16043) and a human AD brain data (PHB) (Accession number: GSE109887, Platform number: GPL10904) (Supplementary Fig. 4a, b). Interestingly, our own transcriptome data had dramatically high GO similar to PHB (CC, 64.9% for PiB iCOs and 76.5% for E4 iCOs; BP, 58.7% for PiB iCOs and 26.0% for E4 iCOs; MF, 46.7% for PiB iCOs and 29.4% for E4 iCOs), whereas PIN had low GO similar to PHB (CC, 7.0% for PIN; BP, 4.2% for PIN; MF, 8.3% for PIN) (Supplementary Fig. 4c). In addition, most of the GO terms in Fig. 2i were also included in PHB (21/28, 75%) as significant terms, but were not included in PIN (6/28, 21%) (Supplementary Fig. 4d). Together, our results confirm that our iCOs had pathological features of AD, and thus could be an appropriate model reflecting characteristics of the actual disease-related human brain lesions.

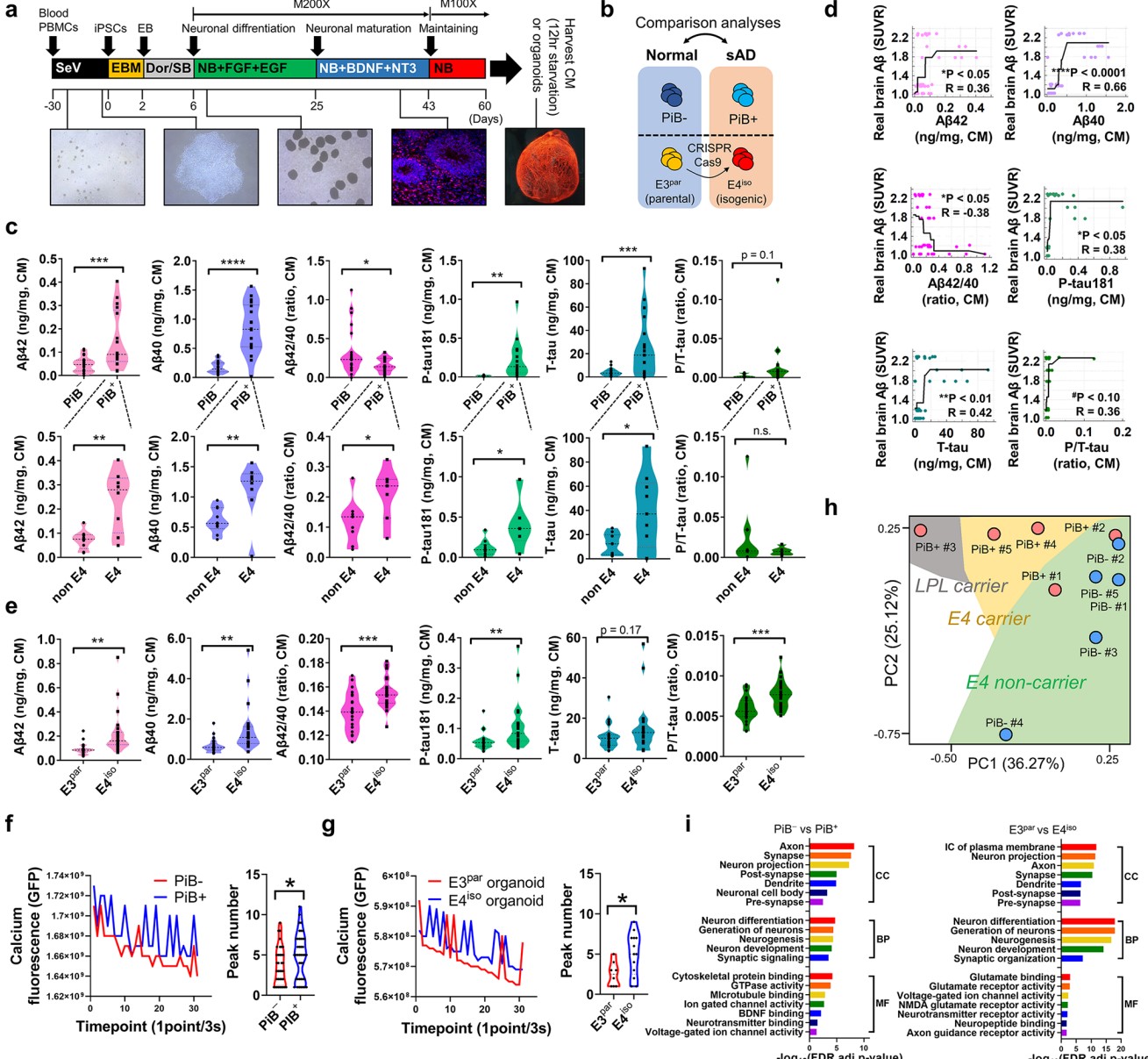

**Fig. 2 Generation of sAD iCOs and their pathological features. a** Generation of iCOs; 60 days-old iCOs were used for the experiments. **b** Experimental iCOs' sets for this study; The iCOs were from both Pittsburgh compound B (PiB)-PET negative/positive participants' iPSCs and ApoE3 (parental) /E4 (isogenic) iPSCs, which were generated by CRISPR-Cas9 gene-editing. **c** Quantification of AD hallmark proteins (Aβ1-42, Aβ1-40, total tau, p-tau) in the organoid conditioned media was performed between PiB⁻ iCOs and PiB⁺ iCOs (***$p = 0.0003$ for Aβ1-42, ****$p < 0.0001$ for Aβ1-40, *$p = 0.0384$ for Aβ1-42/1-40 ratio, **$p = 0.0042$ for p-tau, ***$p = 0.0002$ for t-tau, $p = 0.1025$ for p/t-tau ratio; two-sided $p$-values; unpaired $t$-test; Each 3-7 iCOs was used from $n = 8$ participants) or between non-E4 carriers and E4 carriers (**$p = 0.0020$ for Aβ1-42, **$p = 0.0082$ for Aβ1-40, *$p = 0.0370$ for Aβ1-42/1-40 ratio, *$p = 0.0125$ for p-tau, *$p = 0.0173$ for t-tau, $p = 0.3296$ for p/t-tau ratio; two-sided $p$-values; unpaired $t$-test; Each 3-7 iCOs was used from $n = 4$ participants). **d** Significant correlation was shown between the real human cerebral amyloid deposition (SUVR) and secreted AD hallmark proteins (*$p = 0.0190$ for Aβ1-42, ****$p < 0.0001$ for Aβ1-40, *$p = 0.0183$ for Aβ1-42/1-40 ratio, *$p = 0.0407$ for p-tau, **$p = 0.0042$ for t-tau, #$p = 0.0550$ for p/t-tau ratio; Pearson's correlation; isotonic regression curve was shown). **e** Quantification of AD hallmark proteins (Aβ1-42, Aβ1-40, total tau, p-tau) in the organoid conditioned media was performed between E4iso iCOs and E3par iCOs (**$p = 0.0013$ for Aβ1-42, **$p = 0.0034$ for Aβ1-40, ***$p = 0.0007$ for Aβ1-42/1-40 ratio, **$p = 0.0094$ for p-tau, $p = 0.1720$ for t-tau, ***$p = 0.0003$ for p/t-tau ratio; two-sided $p$-values; unpaired $t$-test; Each 24 iCOs was used from E4iso iCOs and E3par iCOs). **f, g** Comparison of physiological responses of the iCOs (calcium oscillation analysis) was performed between PiB⁻ iCOs and PiB⁺ iCOs or E4iso iCOs and E3par iCOs. PiB⁺ iCOs and E4iso iCOs had more number of peaks than PiB⁻ iCOs and E3par iCOs (*$p = 0.0399$ for PiB⁻ iCOs vs PiB⁺ iCOs, *$p = 0.0254$ for E4iso iCOs vs E3par iCOs; two-sided $p$-values; unpaired $t$-test). **h** Principal component analysis (PCA) plot showing transcriptomic expression patterns in RNA sequencing data. **i** Transcriptomic GO analyses between the PiB⁻ iCOs and PiB⁺ iCOs or E4iso iCOs and E3par iCOs were performed with the FDR-adjusted $p$-value < 0.05 (adjustments were made for multiple comparisons; FDR-corrected by Toppgene analysis). $p$-value criteria: *$p < 0.05$, **$p < 0.01$, ***$p < 0.001$, and ****$p < 0.0001$, two-sided $p$-values, unpaired $t$-test. PBMC peripheral blood mononuclear cells, SeV Sendai virus, EB embryoid body, Dor dorsomorphin, SB SB431542, NB neurobasal media, PiB Pittsburgh compound B, CM culture media, sAD sporadic AD, SUVR standardized uptake value ratio, Aβ beta-amyloid, p-tau phosphorylated tau, MF molecular function, BP biological process, CC cellular component.

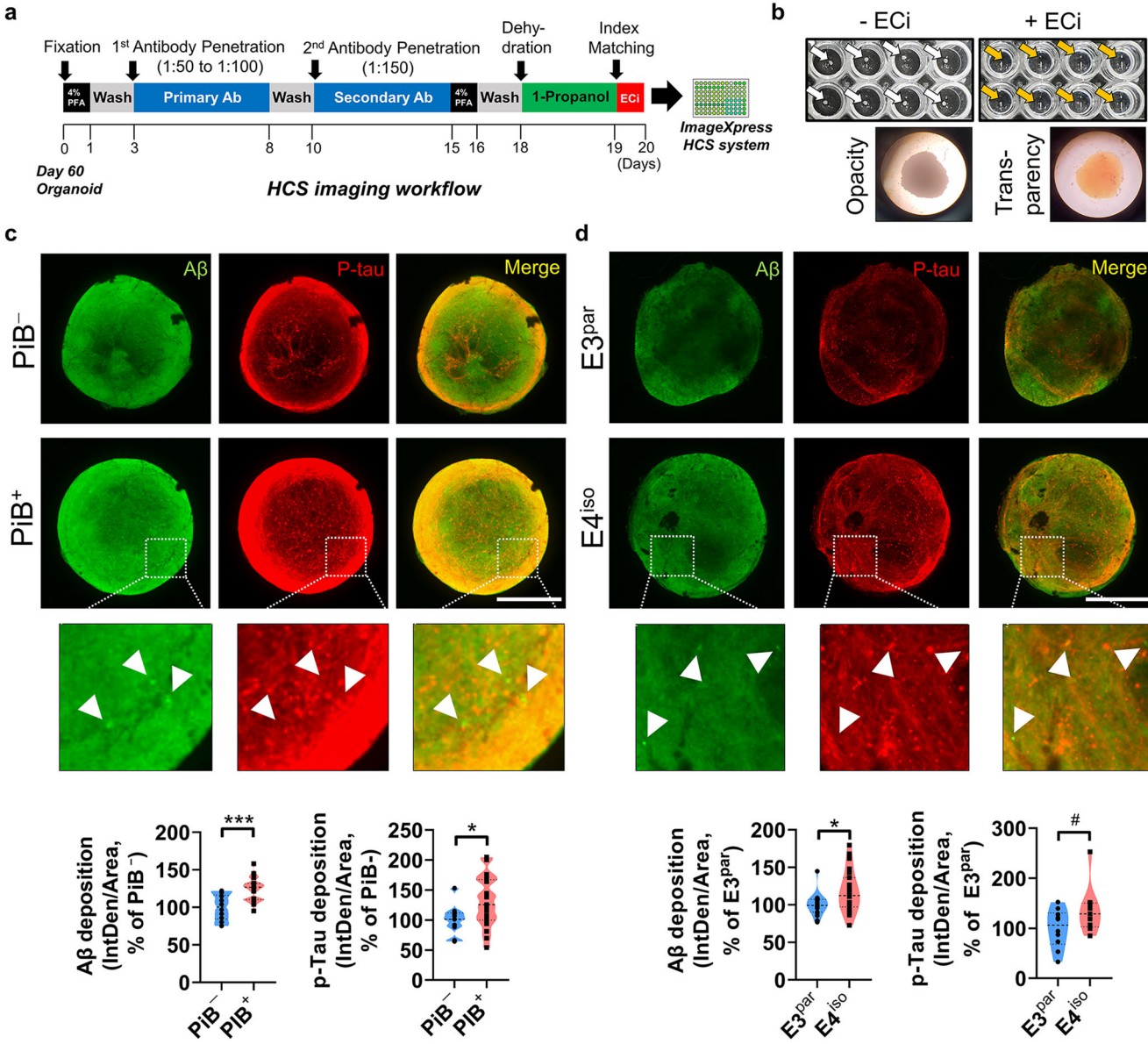

**Fig. 3 Ethyl cinnamate (ECi)-based iCO clearing and ready for the HCS system using iCOs. a** ECi-based iCO clearing and HCS imaging workflow. **b** Transparency of iCOs after the ECi tissue clearing. **c, d** Representative 3D organoid imaging using HCS system and 3D imaging technique after ECi-based tissue clearing procedure. Aβ (red) and p-tau (green) antibodies were used (scale bar, 1 mm) (p-value criteria: #p < 0.1, *p < 0.05, **p < 0.01, and ***p < 0.001; for PiB− iCOs and PiB+ iCOs, ***p = 0.0007 for Aβ, *p = 0.0474 for p-tau; for E4iso iCOs and E3par iCOs, *p = 0.0135 for Aβ, #p = 0.0841 for p-tau; two-sided p-values; unpaired t-test; 10-20 iCOs were used for each group). White arrows, position of Aβ aggregates in the iCOs; White scale bar, 0.5 mm. PiB Pittsburgh compound B, Aβ beta-amyloid, p-tau phosphorylated tau, IntDen integrated density.

**Pathology validation of iCOs using HCS system**. Since we had only checked the levels of secreted proteins in the conditioned media in Fig. 2, we need to examine whether iCOs could exhibit pathological lesions reminiscent of those of real human brain tissues. We used 3D tissue clearing method to create a uniform index following a protocol from Vienna Biocenter (VBC) using ethyl cinnamate (ECi) as an index-matching solution[26] (Fig. 3a). After tissue clearing, iCOs exhibited transparency and were invisible to the naked eye (Fig. 3b). These procedures were performed on iCOs of relatively consistent size plated to 96-well plates, to enable HCS imaging. These methods are described in detail in the Methods. We checked the levels of Aβ and p-tau, and found that they were significantly increased in PiB+ iCOs and E4iso iCOs compared to PiB− iCOs and E3par iCOs (Fig. 3c, d). The utilized Aβ antibody (D54D2) captures all isoforms of

Aβ, and thus the signals were relatively blurry; however, we could detect several aggregated Aβ forms and observed that there was almost no colocalization of the Aβ aggregates with p-tau deposition (Fig. 3c, d, white arrows). As expected, Aβ aggregates generally exist in extracellular regions, whereas p-tau deposition is observed in intracellular regions[27]. Localization of the Aβ plaques or tau tangles was further validated by higher resolution (40X) confocal microscopy imaging (Supplementary Fig. 5). The images clearly show that amyloid-beta aggregates are formed in extracellular interstitial spaces, and hyper-phosphorylated tau colocalizes intracellularly along with neuronal marker MAP2. From these results, we conclude that our iCOs can undergo effective tissue clearing and HCS imaging, and thus could be applicable to the large-scale drug-screening platform.

**Construction of the molecular regulatory network model for AD**. In order to understand the complex dynamics of molecular interactions in our iCOs with pathological features of AD, we need to construct a molecular regulatory network model using systems biology approach. For example, the MAPK signaling pathway is activated in carriers of the ApoE ε4 subtype, which is reflected by increases in Aβ production[28] and CREB expression, the latter of which prevents synapse loss[29]. The canonical WNT signaling pathway delivers an inhibitory signal for GSK-3β, which is one of the major kinases responsible for phosphorylating tau; for ApoE ε4 subtype, this signaling is decreased by the internalization of LRP6 from the membrane[30]. Activation of the non-canonical WNT signaling pathway increases apoptosis, Aβ production, cholesterol production, and the increased autophagy mediated through JNK and RhoA-ROCK[30,31]. Activation of the PI3K-AKT signaling pathway by Reelin and synaptic NMDAR (N-methyl-D-aspartate) suppresses tau hyperphosphorylation and apoptotic signaling. Meanwhile, AKT activates mTORC1 to inhibit autophagy, such that activated AKT subsequently causes accumulation of Aβ[32–34]. In this way, numerous signaling pathways are involved in the regulatory process of AD through complicated interactions. Such complexity makes it difficult to intuitively understand how perturbing a given gene or protein will affect the accumulation of pathological processes.

We herein developed a relevant mathematical model of the neuronal molecular regulatory network for AD, with the goal of enabling researchers to gain a better mechanistic understanding of AD pathological dynamics at a molecular-regulation level and systematically investigate candidate molecular targets for their ability to alter the levels of pathogenic proteins. The procedure of constructing our network model is described in the Methods. In our network model, the neuronal intra-cellular molecular pathways are mainly composed of MAPK signaling pathway, WNT signaling pathway, and PI3K-AKT signaling pathway (Fig. 4a). Genes and proteins are represented as nodes, and interactions between nodes are represented as activation or inhibitory links depending on their type of regulation. Network nodes in our network model and corresponding Boolean logical rules that govern the state of the nodes are given in Supplementary Data 1. Our network model includes five output nodes that can represent the pathological phenotype of AD such as Aβ, p-tau, synapse loss, apoptosis, and autophagy. Aβ and p-tau node activity refer to the levels of these pathological proteins, while synapse loss, apoptosis, and autophagy refer to the degrees of these pathological phenomena. This network model assumes a normal aging state when no input is applied.

To validate whether the constructed network model properly represents the dynamics of AD pathological phenomena, we performed simulations with different levels of oxidative stress, mimicking the aging effect. From the simulation results, we can confirm that our network model can properly reproduce the pathological input–output relationships[35] (Fig. 4b, Supplementary Fig. 7, Supplementary Methods). For experimental validation, the list of altered pathways and their tendencies to increase or decrease was compared[36–38] (Supplementary Figs. 8 and 9). The way of comparing experimental data and simulation results is explained schematically in Supplementary Fig. 8. In addition, to validate the specific allele-related alterations of genes and proteins relative to normal aging, we compared the experimental literature-based knowledge with our simulation results (Supplementary Table 2). Finally, we completed the construction and validation of the molecular regulatory network model for AD.

**Analysis of the AD network model and identification of candidate drugs**. We conducted in silico perturbation analysis to

understand the dynamic behavior of network models for ApoE ε4 allele (E4^iso iCOs) and LPL SNP (LPL^A288T SNP), which showed different pathological features in the previous experiments, and to identify the regulation of candidate targets which can reduce the abundance of pathological proteins.

In the presence of the ApoE ε4 allele, the canonical WNT pathway and synaptic NMDAR signaling are down-regulated while the MAPK pathway is up-regulated which results in increase of Aβ production through increased APP (amyloid precursor protein) expression. Moreover, the autophagy-related signaling pathway is down-regulated through suppressed TFEB, leading to the accumulation of Aβ and p-tau. The non-canonical WNT pathway is up-regulated by increased Aβ, which forms a vicious cycle that increases Aβ production through BACE1 activation. These are consistent with previous biological observations[39–41]. In addition, the increased kinase activity of tau phosphorylation through the up-regulation of Dkk1 and decreased AKT by Aβ leads to the increase in p-tau production and apoptosis signal (Supplementary Fig. 10a). In the case of LPL SNP (reflected as a loss-of-function), the elevated level of cholesterol results in the increase of Aβ production through down-regulated activity of α-secretase and up-regulated BACE1 activity and these alterations were observed in biological experiment[42]. Furthermore, up-regulated mTORC1 activity results in the accumulation of Aβ by suppressing autophagy.

In order to identify the optimal candidate targets for lowering the abundance of Aβ and p-tau, we performed in silico perturbation analysis (Fig. 5a). For this purpose, we performed both single-node perturbation pinning only one node and double-node perturbation pinning two nodes at the same time. In the attractor landscape obtained after arbitrarily fixing one node state set to '0', the average activity in the attractor is multiplied by the ratio of the basin size of the attractor to represent the node activity (Methods). The activity of the output nodes was converted to a phenotype score representing the degree of proximity to the desired state based on the assigned weights according to the priority of importance for reducing the abundant levels of Aβ, p-tau, and the degree of neurodegeneration (i.e., synapse loss and apoptosis).

From the simulation results, we selected those targets that have high phenotype scores as far as FDA-approved drugs are available for drug repositioning to inhibit them, and further analyzed the alteration of signaling pathways by perturbation of single targets or double-target combinations (Fig. 5). For instance, in the case of ApoE ε4 allele, the treatment with Flibanserin, an inhibitor of PTEN, and Ripasudil, an inhibitor of Dkk1, up-regulates the canonical WNT pathway and down-regulates the non-canonical WNT pathway (Fig. 5b, top and left). The alteration of these pathways subsequently decreases the production of Aβ owing to the increase in α-secretase activity and the decrease in BACE1 activity. In addition, suppression of PTEN inhibits apoptotic signaling and down-regulates GSK-3β activity through the activation of AKT (Supplementary Fig. 10b), which consequently inhibits the production of p-tau. As another example, when PTEN and mTOR are turned OFF by the treatments with Flibanserin and Everolimus, respectively, the production of both Aβ and p-tau is decreased by the elevation of autophagy (Fig. 5b, top and right), which reflects the mTOR OFF effect and the aforementioned PTEN OFF effect. As a result, the activities of two output nodes, Aβ and p-tau that represent pathological proteins, are decreased for both combinatorial treatments. Taken together, we suggest that the double-target treatment would be needed for the case of the ApoE ε4 allele since there is no single target that can simultaneously inhibit Aβ production and synapse loss, and also up-regulate autophagy. In the case of LPL SNP, the activity of α-secretase increased by the Ripasudil-triggered suppression of

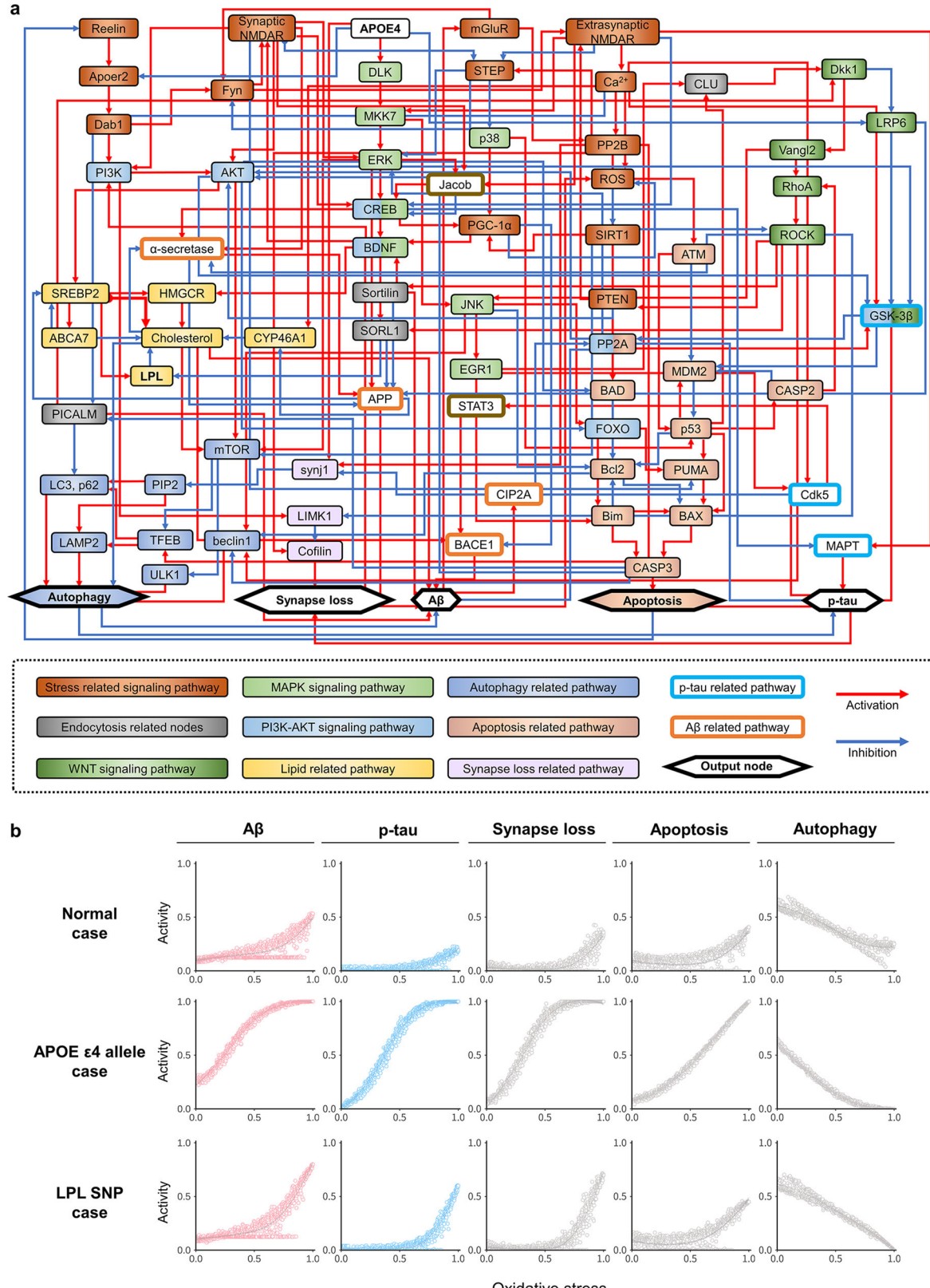

**Fig. 4 The molecular regulatory network model of AD and its validation. a** The molecular regulatory network model of AD was constructed with 77 nodes and 204 regulatory links. The colors of nodes indicate the corresponding KEGG pathways. **b** Simulation response profiles of the network models (*n* = 5) to oxidative stress by varying oxidative stress (ROS) from 0% to 100% over one thousand (*n* > 1000) independent simulations for each condition (normal, ApoE ε4 allele (APOE4) case, LPL SNP case). Gray solid lines denote 95% confidence intervals around the mean value and each data-point means independent simulations (error band is so narrow and mostly not distinguishable from the line of mean value). Aβ beta-amyloid, p-tau phosphorylated tau, APOE apolipoprotein E, LPL lipoprotein lipase.

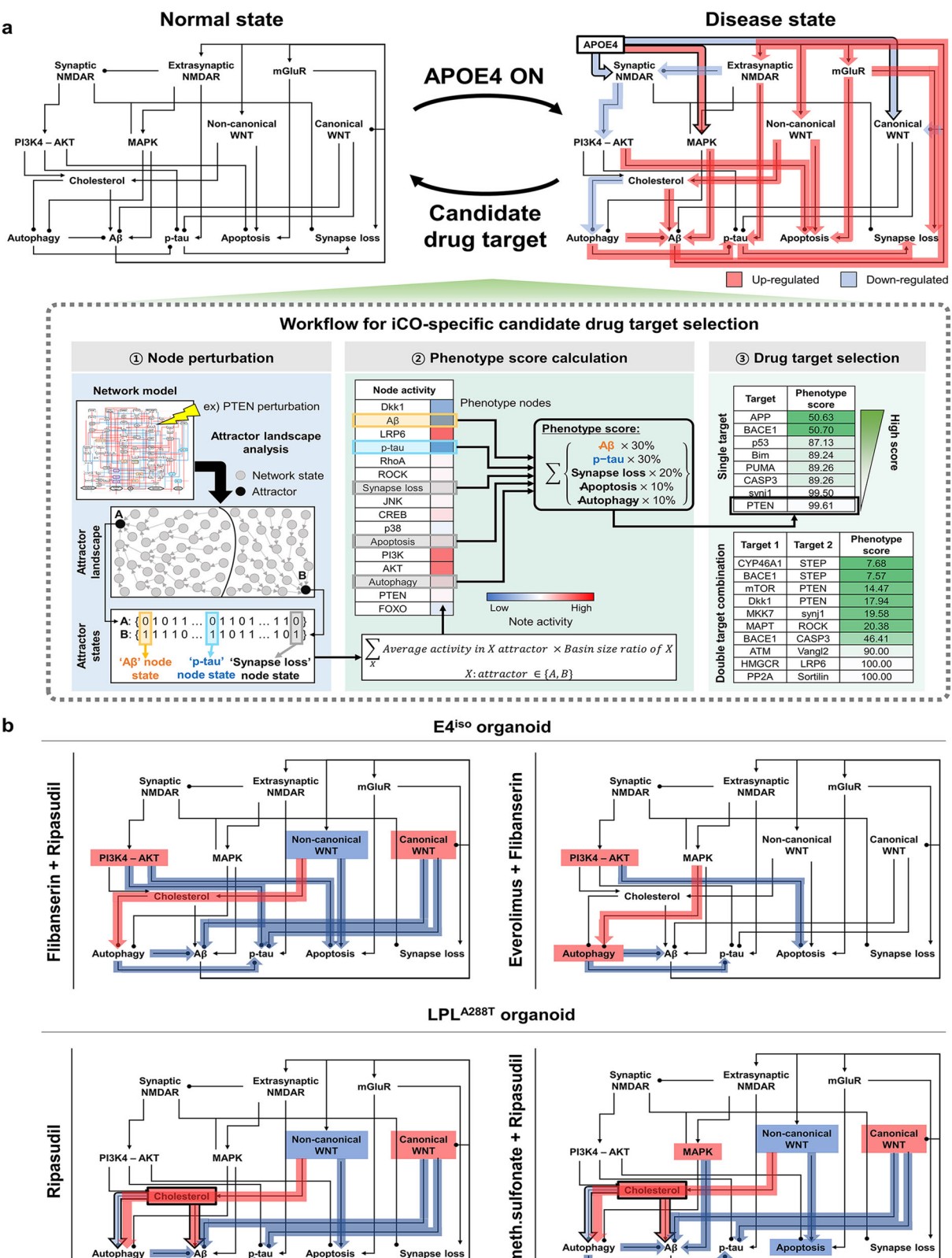

the non-canonical WNT pathway and results in the decreased production of Aβ, whereas the Ripasudil-triggered elevation of AKT activity suppresses apoptotic signaling (Supplementary Fig. 10c).

In sum, we could propose candidate drugs based on systems analysis of the dynamical network model with detailed regulatory mechanisms. The original usage of the FDA-approved drugs that were suggested from this analysis and expected to be targeted in sAD iCOs are listed in Supplementary Table 3.

**Validation of our network-based drug-screening platform using iCOs and HCS.** The selection of candidate drugs was done

**Fig. 5 Perturbation analysis-based selection of candidate drug targets using attractor landscape analysis. a** Workflow for risk factor (ApoE ε4 allele and LPL SNP)-specific candidate drug target selection. Disease models, such as ApoE ε4 allele case and LPL SNP case, are represented by differentially wired networks that have a distinct network topology by mapping their risk factors in a disease model onto the normal state model. Network dynamics induced by a node perturbation can be analyzed by an attractor landscape which consists of the trajectories from $10^6$ initial states to the attractor states. Each node perturbation eventually reaches the attractor states that correspond to specific cellular phenotypes. The area around each attractor state is the region of states with trajectories converging to the attractor, which is called the "basin of attraction" or "basins", and can be used for measuring the relative stability of the specific cellular phenotypes, including Aβ, p-tau, apoptosis, synapse loss, and autophagy. The overall cellular state for a specific node perturbation is defined by a phenotype score which measures the sum of products that multiply the basin ratio of attractors belonging to same cellular phenotype and the distinct weight corresponding to the specific cellular phenotype (Aβ:30, p-tau:30, synapse loss:20, apoptosis:10, autophagy:10). The phenotype score ranges from 0 to 100 and is used to estimate the pathological level. **b** Schematic diagram of the predicted perturbation effect of the high-rank FDA-approved drugs on the activities of signaling pathways for ApoE ε4 allele case (upper row) and LPL SNP case (lower row). The color of each pathway represents its activity change by drug treatment (i.e., red for induction and blue for suppression), and the color of arrow represents the pathway effect on phenotypes. APOE apolipoprotein E, iCO iPSC-derived cerebral organoid, Aβ beta-amyloid, p-tau phosphorylated tau, LPL lipoprotein lipase.

according to the following steps (Fig. 6a): (i) output node priority selection; (ii) target drug selection based on perturbation analysis and reference to a library of FDA-approved drugs; and (iii) exclusion of unsuitable candidates based on their drug properties (BBB penetrability, carcinogen status, etc.). The finally selected candidates are listed in Fig. 6a. The drugs mainly targeted E4[iso] iCOs and PiB[+] iCOs with ApoE e4 allele (considering ApoE-related pathways); some targeted PiB[+] #3 iCOs, which had the LPL[A288T] SNP (considering LPL-related pathways).

Given the many reports that iCOs exhibit size variations during their growth, we next sought to minimize the possible variation to improve their potential utility as drug-treatment targets. We performed three quality control (QC) steps, and obtained well-shaped and evenly sized iCOs (Supplementary Fig. 6 and Fig. 6b). The details of the utilized QC protocol are described in the Methods. For drug screening, we treated the iCOs with the selected drugs and monitored the levels of Aβ or tau deposition in a manner similar to the experiments presented in Fig. 3. We found that all of the candidate drugs were effective to some degree in reducing Aβ or tau deposition and in enhancing or maintaining neuronal cell viability (Fig. 6c–e, Supplementary Fig. 11). These results, which are summarized in Supplementary Fig. 12, indicate that we successfully validated our network-based drug-screening platform by integrating mathematical modeling and pathological traits of human iCOs. We thus herein introduce a reliable strategy that could enable precision medicine by engaging the convergence of mathematical modeling and pathological features of brain organoids.

## Discussion

In this paper, we developed a drug-screening platform and propose a strategy for the precision medicine by integrating mathematical modeling and human iCOs. Although there are studies that applied mathematical modeling to AD[43,44], no study has attempted to combine mathematical modeling with human iCOs that express pathological features of AD. Please note that our iCOs fully represent sAD conditions because we generated iCOs from various perspectives, including PiB[−] iCOs without ApoE ε4 allele, PiB[+] iCOs with or without ApoE ε4 allele, and CRISPR-Cas9-edited apolipoprotein E (ApoE) ε4 isogenic iPSC lines (E3[par] and E4[iso] iCOs) (Fig. 1). Also note that we used a large number (~1300) of iCO samples in this study, which was sufficient to identify pathological phenotypes and drug responses.

We had to consider the following critical points when establishing our drug screening model. First, although numerous studies have already shown the possibility of mechanism-based understanding and control target discovery through dynamical modeling[45–50] for cancer cells, there were no mathematical models for the molecular regulatory interactions in the neuron, which also have complex dynamics that are difficult to intuitively

understand[51,52]. Therefore, to understand the functional role of each molecular component and identify mechanism-based control targets, we needed to investigate the interactions of the components within the interaction network considering the dynamics of the molecular network. For these reasons, we developed and analyzed the neuronal molecular regulatory network and presented the mathematical model of a molecular regulatory network considering dynamics in the neuron made by integrating all available experimental evidence. Second, many researchers have claimed that the homogeneity of testing samples is important for a highly controlled drug-screening platform[53,54]. Since there have been many reports that point out the sample-to-sample variability of human brain organoids[55–58], especially on their size variations[58], the first thing we focused was the way to control the quality of our iCOs. We performed several steps of QCs and finally got well-shaped and even-sized iCOs. As shown in Supplementary Fig. 6, our iCOs had uniform shape and size, and were optimized for the drug-screening platform. It is meaningful that our drug screening system suggests a possibility to utilize iCOs as drug-treatment targets, away from the existing drug-screening platforms that use only 2D neurons or small 3D neurospheres (diameter: <300 μm) derived from neural stem cells (NSCs)[59,60]. Although one paper showed the HCS system using human iCOs, they did not check pathological hallmark proteins such as Aβ or tau deposition or focus on the AD[61]. Finally, we tried to apply FDA-approved drugs on our drug screening model to show the possibilities of drug repositioning and simplify the drug approval process in preparation for the practical use (Fig. 6). We narrowed down the candidate drugs from the list of FDA-approved Drug Library Plus provided by the MedChemExpress (MCE) company. Although they were not initially developed for AD, their mechanisms were clearly linked with our mathematical model's pathways. Therefore, we could speculate how the drugs would show effectiveness in reducing Aβ or tau deposition and in enhancing neuronal cell viability. We found that these drugs are expected to have effects on AD-related pathways which were shown in Supplementary Table 3. Thus, we suggested that our drug screening system is a technologically advanced platform with highly controlled mathematical model and thoroughly validated samples.

Our current study has several limitations worth noting. This network model assumes an initial state with little neuronal loss. Therefore, the adjustment of network model by disease stage is necessary. In addition, since it is a network model that consist of limited and only observable experimental information, considering the quality of data that will be developed in the future, it will be possible to create a more complex, emergent decision network that can be analyzed in an advanced manner. Next, even though we generated iCOs from iPSCs as a biomimetic mini-brain, microglial population cannot emerge embryologically

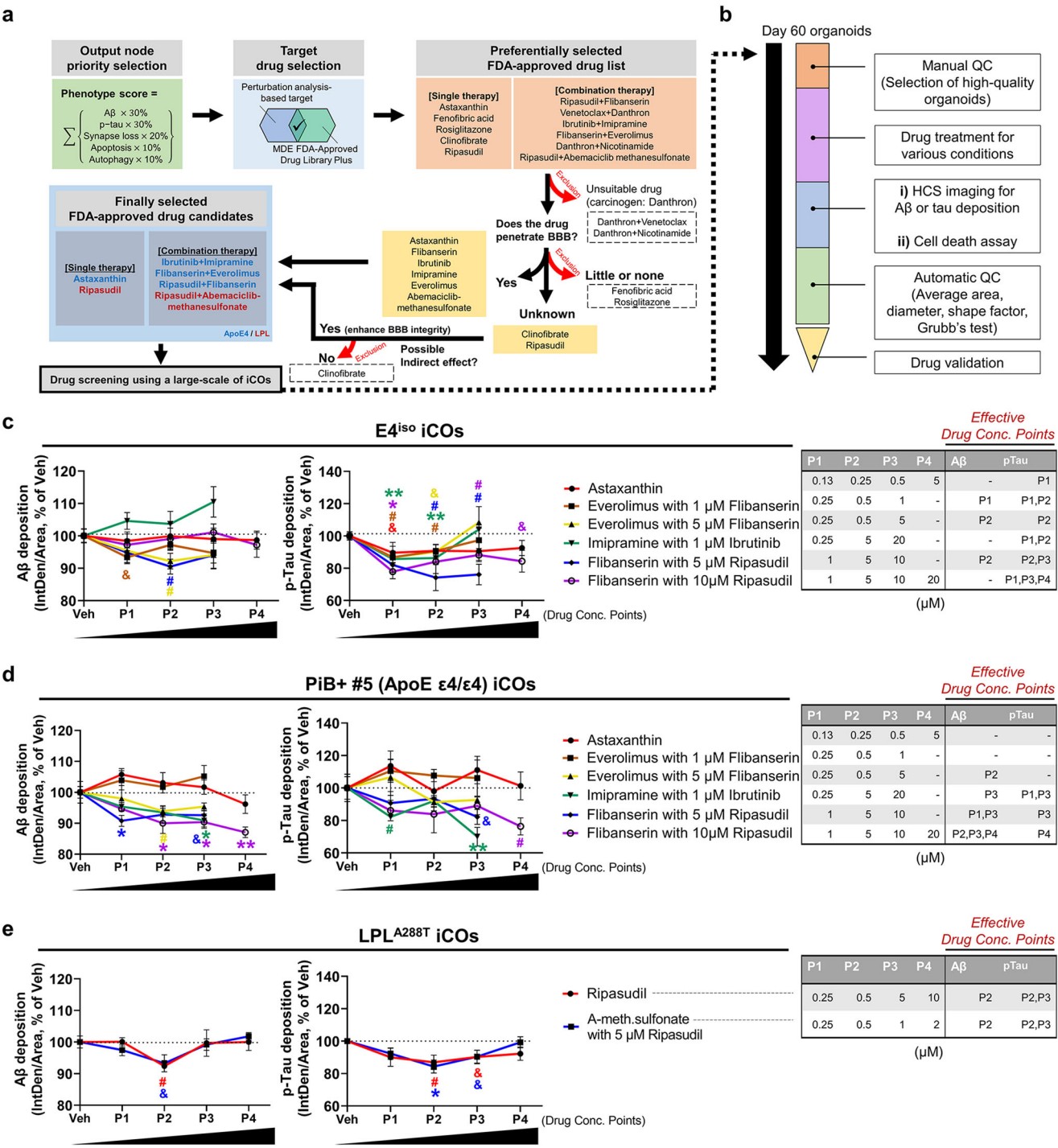

during the developmental process of iCOs because their origin has been known as yolk-sac while it is a matter of debate[62]. Since microglia have also critical roles in the immune responses of the human brain[63], our drug screening model was unable to deal with inflammation-related drug responses. Further research through mixed-culture of iPSC-derived microglia and iCOs could help us find a way to create more accurate drug screening system combined with the mathematical modeling. Moreover, we could further proceed to mechanistic studies using large quantities of our iCOs to verify the specific pathways associated with the drug candidates, as well as quantitative identification of Aβ/tau deposition or neuronal cell death. In particular, canonical and

non-canonical WNT pathway and autophagy-related pathways were signified in our mathematical model. Therefore, these pathway-molecules such as PTEN, Dkk1, RhoA, ROCK (related to non-canonical WNT pathway) and mTOR, ULK1 (related to autophagy pathway) have to be further validated.

In conclusion, we provide a powerful tool for the development of the AD drugs by our network-based drug-screening platform through the integration of mathematical modeling and the pathological features of human iCOs, which are derived from the sporadic AD patients. With further generation of the iCOs from various types of AD patients, our approach may propose strategies for the precision medicine therapy.

**Fig. 6 Validation of our network-based drug-screening platform. a** The flow-chart of the drug selection steps; six FDA-approved single or combination drugs were selected (for E4$^{iso}$ iCOs: Astaxanthin, Ibrutinib+Imipramine, Flibanserin+Everolimus, Ripasudil+Flibanserin; for LPL$^{mut}$ iCOs: Ripasudil, Ripasudil+Abemaciclib-methanesulfonate) and five inappropriate therapies (Danthron + Venetoclax and Danthron+Nicotinamide, possible carcinogen; Fenofibric acid and Rosiglitazone, poor BBB penetration rate; Clinofibrate, unknown BBB penetration) were excluded from the list. It is also not known whether Ripasudil is BBB permeable, however, it is included as a candidate because of its possible role in brain health (enhance BBB integrity). **b** The QC steps of iCOs. The high-quality iCOs were manually or automatically selected and used for the drug screening. **c-e** A large-scale drug screening results from our network-based drug-screening platform. E4$^{iso}$ iCOs, PiB$^+$ #5 iCOs, and LPL$^{A288T}$ iCOs (PiB$^+$ #3) were used for the screening. $p$-value criteria: $^{\&}p < 0.1$ for unpaired $t$-test (two-sided $p$-values), $^{\#}p < 0.05$ for unpaired $t$-test (two-sided $p$-values), $^{*}p < 0.05$ and $^{**}p < 0.01$, ANOVA with the correction of Tukey post hoc test for multiple comparisons. For Aβ of E4$^{iso}$ iCOs, $^{\&}p = 0.0578$, $^{\#}p = 0.0471$, $^{\#}p = 0.0109$; For p-tau of E4$^{iso}$ iCOs, $^{**}p = 0.0065$, $^{*}p = 0.0306$, $^{\#}p = 0.0120$, $^{\&}p = 0.0594$, $^{\&}p = 0.0776$, $^{\#}p = 0.0157$, $^{**}p = 0.0089$, $^{\#}p = 0.0140$, $^{\#}p = 0.0449$, $^{\#}p = 0.0111$, $^{\&}p = 0.0622$; For Aβ of PiB$^+$ #5 iCOs, $^{*}p = 0.0440$, $^{\#}p = 0.0491$, $^{*}p = 0.0342$, $^{\&}p = 0.0664$, $^{*}p = 0.0232$, $^{*}p = 0.0234$, $^{**}p = 0.0021$; For p-tau of PiB$^+$ #5 iCOs, $^{\#}p = 0.0480$, $^{\&}p = 0.0842$, $^{**}p = 0.0043$, $^{\#}p = 0.0365$; For Aβ of LPL$^{A288T}$ iCOs, $^{\#}p = 0.0180$, $^{\&}p = 0.0831$; For p-tau of LPL$^{A288T}$ iCOs, $^{\#}p = 0.0347$, $^{*}p = 0.0164$, $^{\&}p = 0.0512$, $^{\&}p = 0.0546$ (in order from left to right, top to bottom). Each color indicates different types of candidate drug. Effective drug concentration points (P1 to P4) are shown in tables. $n = 6$ iCOs were used for each concentration point (P1 to P4). Data are presented as mean values ± standard error of mean (SEM). APOE apolipoprotein E, iCO iPSC-derived cerebral organoid, Aβ beta-amyloid, p-tau phosphorylated tau, LPL lipoprotein lipase, QC quality control, Conc. concentration, A-meth.sulfonate Abemaciclib methanesulfonate.

## Methods

**Recruitment of participants and brain amyloid imaging**. For the generation of induced pluripotent stem cells (iPSCs) and iCOs, 10 participants were recruited. Detailed information on the participants is presented in Supplementary Table 1. Each participant underwent simultaneous 3D $^{11}$C-Pittsburgh compound B positron emission tomography (PiB-PET) and MRI using a 3.0 T PET-MR scanner (Siemens Healthineers) for brain amyloid imaging. The methods used to obtain and process the patients' imaging data are described in our previous paper, in which the characteristics of our recruited participants were further summarized[4,64].

**Human iPSCs and iCOs**. Detailed methods for generation, maintenance, and characterization of both iPSCs and iCOs are provided in the Supplementary Methods.

**Ethyl-cinnamate (ECi) 3D tissue clearing**. We followed a previously published protocol[26] with minor adjustments for use with a 96-well plate. Detailed methods for ECi 3D tissue clearing for iCOs are provided in the Supplementary Methods.

**High-content screening 3D confocal image acquisition and analysis**. Images were acquired using an ImageXpress Micro Confocal High-Content Imaging System (Molecular Devices) with a 4X Plan Apo objective. A total of 25 planes were acquired in 50-μm intervals that covered the whole organoid. All images and their maximum projection images were used for quantification of fluorescence. Images were analyzed using the MetaXpress High-Content Image Acquisition and Analysis Software (Molecular Devices). The following organoid modules were used to characterize each sample: (i) Source, Open Close with the circle filter shape; (ii) Adaptive Threshold: TexasRed Segmentation; and (iii) approximate width from 300 to 3000 μm. Samples that did not meet the predefined standards were excluded from further analyses. The iCOs were further quality-controlled for HCS acquisition and analysis (see below).

**Drug treatment**. For drug treatment, detailed information on concentrations is shown in Fig. 6 and is provided in Supplementary Methods.

**Quality controls of iCOs for drug screening**. First, high-quality embryoid bodies (EBs) were selected on Day 7 and seeded to the ultra-low-attachment 96-well plates. Second, on Day 60, high-quality (less size-variation) iCOs were selected and re-seeded. Third, during the HCS imaging, automatic QCs were performed with the following exclusion criteria: (i) shape factor score <0.7; (ii) diameter length <1000 μm or >1400 μm; and (iii) average area of iCOs < 700,000 μm$^2$. Supplementary Fig. 6 shows these QC steps.

**Immunocytochemistry and immunohistochemistry**. Detailed methods for immunocytochemistry and immunohistochemistry for both iPSCs and iCOs are provided in the Supplementary Methods.

**Cell viability assay**. To check cell viability for iCOs, we performed 3-(4,5-dimethylthiazol-2-yl)-2,5-diphenyltetrazolium bromide (MTT) assay for iCOs with minor modification of the method from our previous report[65]. In detail, 0.9 mg/ml of MTT (475989, Sigma) in opti-MEM was treated and the plates were incubated for 2 h at 37 °C. After that, medium was fully removed and isopropanol was added and incubated again for 2 h at 37 °C. When formazan crystals were dissolved in the isopropanol solution, absorbance was measured at 540 nm.

**Quantification of secreted protein levels in the conditioned medium**. To analyze protein secretion under the serum-deprived condition, the culture medium was changed to Opti-MEM and samples were incubated at 37 °C under 5% $CO_2$ for 12 h. The culture medium was collected and centrifuged at 3000$g$ for 10 min at 4 °C, and the supernatant was collected and stored at −80 °C. The proteins levels of Aβ1-40 (27713, IBL), Aβ1-42 (27711, IBL), phospho-tau (pT181) (KHO0631, Invitrogen), and total-tau (KHB0041, Invitrogen) were measured by enzyme-linked immunosorbent assay (ELISA) according to manufacturer's instructions. BCA analysis was performed with the same samples, and the data were normalized by the total protein content. Levels of Aβs secreted from E3$^{par}$ and E4$^{iso}$ iCOs were further measured using $xMAP$ technology (Bioplex 200 systems). The utilized protocol is also described in our previous paper[66].

**Reverse transcriptase quantitative PCR (RT-qPCR)**. Detailed methods for RT-qPCR are provided in the Supplementary Methods.

**Calcium oscillation analysis with the FLIPR calcium 6 assay**. Detailed methods for FLIPR Calcium 6 assay are provided in the Supplementary Methods.

**RNA sequencing, differentially expressed genes, and gene ontology analysis**. Detailed methods for RNA sequencing, DEGs, and GO analysis are provided in the Supplementary Methods. RNA sequencing data is available at NCBI under SRA accession number PRJNA678865. To analyze public transcriptome data (Accession number: GSE143951, GSE109887; Platform number: GPL16043, GPL10904)[67,68], GEO2R analyzer (https://www.ncbi.nlm.nih.gov/geo/geo2r) was used.

**Construction of Alzheimer's disease molecular regulatory network**. The network structure was constructed based on major signaling pathways related to AD by integrating information from public databases, such as Kyoto Encyclopedia of Genes and Genomes (KEGG)[69] and AlzPathway[70], and also by an extensive survey of relevant experimental data on neuronal cells. In our study, we focused on APOE ε4 allele, LPL-related signaling pathways in line with experimental results of this study. Through a literature-based investigation, we constructed a model network mainly from MAPK, WNT, and mTOR signaling pathways, and also, we considered other signaling pathways such as Notch, RELN, Jak/Stat, and Ca$^{2+}$. In particular, our network model assumes a normal aging state and is constructed in consideration of those pathways relevant for ApoE ε4 allele and LPL. Genes and signaling proteins were represented by nodes, and a physical or chemical interaction between nodes was represented by a link. Our network model consists of 77 nodes and 204 links.

**Mathematical modeling**. We constructed a mathematical model by employing Boolean network modeling[71] and using the regulatory information of the genes and proteins in the context of neuronal cells[72,73]. In our network model, the state values of each node can be either 0 or 1, representing 'OFF' or 'ON' state of the gene/protein activity, respectively. Each node state is updated synchronously according to the logical rule which was established based on experimental data from the literature. We used R package 'BoolNet' for the Boolean simulation[74]. The network state is determined by the set of node states. All the network states eventually converge to a stable state which is called an 'attractor' representing a particular biological phenotype[75,76]. The set of all initial states converging to a particular attractor is called the 'basin of attraction' of the attractor. Due to the high computational cost of simulating all initial states, we randomly sampled one million initial states. We confirmed that the main results were not sensitive to the randomly selected initial states through repeated sampling process. The Boolean functions of our model were written in accordance

with the BoolNet format, and they were converted to SBML-qual[77] format using BioLQM v0.6.1[78]. In addition, the SBML file and R code for simulation are available at: https://doi.org/10.5281/zenodo.4259960.

**Attractor landscape analysis**. Attractor landscape is the landscape of all attractors that can exist on a given network model and the set of initial states converging into each attractor. Using attractor landscape information, we calculate node activities by averaging attractor states weighted by their basin sizes. Each node activity represents the expression level or activity level of gene or protein, respectively. Thus, we can use these node activity calculations to estimate perturbation effects in silico. For example, if A node's activity level is increased when B node is perturbed, we can consider the A node's activity level change as the perturbation effect of the B node perturbation on A node.

**Input–output relationships of the network model**. Detailed methods for input–output relationships for the network model are provided in the Supplementary Methods.

**Statistical analysis of experimental data**. MedCalc 17.2 (MedCalc Software, Ostend, Belgium) and GraphPad Prism 8 (GraphPad Software, CA, USA) were used for data analyses. The numerical data were tested using ANOVA with Tukey's post hoc test or independent $t$-test. The relationship between variables was determined by Pearson's correlation analysis.

**Ethical approval**. Approval for the study was obtained from the Institutional Review Board of Seoul National University Hospital, South Korea. Participants or their legal guardians provided written informed consent.

**Reporting summary**. Further information on research design is available in the Nature Research Reporting Summary linked to this article.

## Data availability

The datasets generated and analyzed are available from the corresponding author upon appropriate request. The RNA sequencing data from this study (related to Figs. 2, 4, 5, and Supplementary Figs. 2, 4, 8, 9) are available at NCBI under SRA accession number PRJNA678865. The public transcriptome data that support the findings of this study (related to Supplementary Fig. 4) are available in GEO2R public database (Accession number: GSE143951 and GSE109887).

## Code availability

The computer codes used to perturbation analysis are available from GitHub (https://doi.org/10.5281/zenodo.4259960).

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

## Acknowledgements

This work was supported by grants from the Korea Health Industry Development Institute (KHIDI), funded by the Ministry of Health & Welfare, Republic of Korea (HI18C0630) for I. Mook-Jung. It was also partially supported by the National Research Foundation of Korea (NRF) grants funded by the Korea Government, the Ministry of Science and ICT (2020R1A2B5B03094920) and the Electronics and Telecommunications Research Institute (ETRI) grants [20ZS1100, Core Technology Research for Self-Improving Integrated Artificial Intelligence System and 20YB1200, In-Silico Brain Model for Assessment and Prediction of Therapeutic Efficacy and Toxicity] for K.-H. Cho, and supported by a grant of the Korea Health Technology R&D Project through the Korea Health Industry Development Institute (KHIDI), funded by the Ministry of Health & Welfare, Republic of Korea (Grant No: HI18C0630 & HI19C0149) for D.Y. Lee. We thank Danbi Jung for technical support for organoid-culture and Jonghoon Lee and Junsoo Kang for their critical reading and comments. Also, we sincerely appreciate for the given ApoE3 iPSC line from Dr. Yankner at Harvard Medical School, and ApoE4 isogenic iPSC line from Dr. Tsai at MIT.

## Author contributions

J-C.P., S.Y.J., D.L., K-H.C., and I.M-J. conceptualized this study. J-C.P., D.L., and H.J.K. carried out experiments. J.L., H.C., and S-H.H. analyzed statistics. M.S.B., D.Y., J.S., and D.Y.L. provided the resources, and S.Y.J., U.K., H.C., and K-H.C. conducted mathematical modeling and simulation analysis. J-C.P., S.Y.J., and D.L. wrote the original draft. K-H.C. and I.M-J. reviewed and edited the manuscript.

## Competing interests

The authors declare no competing interests.
