## [Peer Review File · Nature Communications]

Reviewers' Comments:

Reviewer #1:

Remarks to the Author:

The manuscript by Park et al. describes results from work that integrates mathematical modeling predictions regarding combinatorial drug targets and testing these predictions in iPSC-derived cerebral organoids as a high-content drug screening system.

There are many positive aspects of the manuscript, and overall I am very enthusiastic about the presented results. The manuscript is very well written, easy to follow, and the presented results can be of broad scientific interest. The development of a high-throughput drug screening system that incorporates advanced organoid systems and mathematical modeling is a substantial strength. While I imagine this workflow can test new molecules, inhibitors, etc., its more exciting aspect lies in the ability to screen already approved drugs for new clinical indications quickly, thus reducing the cost and time to new potential treatments.

However, I believe several areas of concern are essential to address to improve the manuscript and interpretation of the results prior to acceptance.

* Research reproducibility of the modeling results: While the Boolean functions of the model are available as plain text in the Supplementary Excel file, the model should be shared in the SBML-qual format (PMID: 24321545) to make it easier for others to reproduce the presented research, and to further expand the model. At a minimum, this SBML file should be available as part of this manuscript. Better would be making the model available in any or all of existing Boolean model repositories (e.g., BioModels (which has been supporting Boolean models for a while now; PMID: 29106614), GINsim.org, Cell Collective; PMID: 22871178).

* Related to reproducibility, I don't believe the methods for the modeling component are comprehensive enough. For example, what initial state(s) did the authors use for the quality control input-output analyses (Fig. 5 and Fig. 5-1), and the actual predictions? Did the authors perform initial condition sensitivity analysis?

* For the oxidative stress input-output analyses: what activity levels were used for other external variables (e.g., Reelin, APOE4, etc.)? Did the authors do these analyses for multiple external conditions?

* It is not clear how oxidative stress ("ROS") in the model was used for input-output analyses with varying activity levels, because the model (Fig 4 and Excell file) shows that several other model components regulate ROS -- in other words, it is not an independent variable (input) that could be easily used for input-output analyses.

* The authors did an excellent job annotating the model with citations specific to the model components' interactions. This level of annotations will aid in the transparency of the model. For readability issues, can the authors replace the PMIDs in all supplementary documents with actual citations and references?

* In addition to providing more details about the modeling methodology and model/simulation set-up, it would be helpful if the authors included citations to offer readers with options further to dive even deeper into Boolean/logical modeling. Examples of more recent reviews on the topic include PMID: 27303434, 32313939, etc. BoolNet should be cited (PMID: 20378558). Finally, the input-output analysis of the model leverages the conversion of the binary input/output of Boolean networks into semi-continuous Activity Levels and "%ON" concepts originally developed in Helikar et al., 2008. As such, it might be appropriate to cite the work.

* Some of the figures are very complex, making them hard to read without zooming in

significantly (Fig 6 is the least readable)

A few questions on the drug effect work/results:

- * For the drug effect studies, can the authors please more clearly explain what was/were the control(s)?
- * How many replicates were used?
- * How did the authors select the drug concentrations?
- * Why did the authors use 24h for the drug treatment? Did the analysis post-treatment happen at the 24-hour mark, or was it later? Why not use 48 or 72 hours?

Minor concerns:

- * Line 45: remove "the" before precision medicine
- * Line 50: remove "the" before symptoms
- * Line 76: "in vitro" should be italicized
- * Line 99: "Mathematic" should be "Mathematical"
- * Line 128: Missing "genes" from "differentially expressed genes (DEG) patterns"
- * Line 197: Remove "comprehensive" -- while the model is not a "toy model," comprehensive is a relative term, and I don't believe this model is comprehensive (for example many other pathways known to interact with the modeled pathways could be included). Perhaps "relevant" is a more appropriate word?
- * Line 313: Similar to above, I recommend removing "the whole" because the model does not include the entire interaction network
- * Line 321: "was focusing" does not seem grammatically correct; perhaps change to "was to focus on.."?
- * Line 333: remove "but"

Best wishes,

Tomas Helikar

Reviewer #2:

Remarks to the Author:

This manuscript entitled "A network-based drug-screening platform for Alzheimer's disease by integrating mathematical modeling and pathological features of human brain organoids" describes a network-based drug-screening platform developed by integrating mathematical modeling and the pathological features of Alzheimer's disease (AD) with human iPSC-derived cerebral organoids. Basic idea of this manuscript is a novel and interesting in developing drug-screening for Alzheimer's disease.

However, authors should clearly describe their methods in construction of AD molecular network. How did they narrow genes/proteins/phenotypes down to only 77 genes/proteins/phenotypes? It appears to be an arbitrary choice. At least, AlzPathway consists of 1,347 species (genes and proteins) and 129 phenotypes. Authors need to explain the reason why they focus on MAPK signaling pathway, WNT signaling pathway, and PI3K-AKT signaling pathway as molecular regulatory network of AD (Fig. 4A) which is a basis for network analysis in this manuscript. By the way, Fig. 4A is the correct figure? For example, Reelin has relationships with not only Apoer2 but also VLDLR and Apoptosis according to a logic table (Supplementary Table 2).

Authors also should clearly describe their methods in analysis of the AD network model and identification of candidate drugs. For example, Fig. 5A illustrates up-regulated and down-regulated pathways according to their perturbation analysis, but they did not explain the definition of "up-regulated" and "down-regulated" pathways, and they did not show their results of perturbation analysis. Authors also should clearly describe their methods of attractor landscape analysis and unfortunately attractor landscape drawn in Figure 5B is too small to see.

By the way, the idea of phenotype score is interesting to estimate the pathological level, but there are several concerns. Authors choose key proteins and phenotypes such as A β , p-tau, synapse loss, apoptosis, and autophagy for calculation of phenotype score, but how to choose these key proteins and phenotypes? Phenotype score looks work well but p-tau proteins have double impact on the score because p-tau activates synapse loss. They developed a network model but they did not consider these kinds of network effects in the calculation of phenotype score. If phenotype score well works to estimate the pathological level, why did they need to develop a network model and conduct a perturbation analysis? Why don't authors directly conduct the phenotype score calculation?

Basic idea of this manuscript is a novel and interesting in developing drug-screening for Alzheimer's disease, but they need to clearly describe their methods and results.

Reviewer #3:

Remarks to the Author:

This is an interesting paper in which Park et al. first generate a cerebral organoid (CO) model of Alzheimer's disease (AD) through two complementary approaches: (1) using patient derived iPSCs (PiB+ iCOs) and (2) using isogenic iPSC lines with a CRISPR-edited apoE ϵ 4 allele (E4iso iCOs).

They do an in-depth characterization of their AD organoids to show that they recapitulate some key pathological phenotypes of the disease, which are absent in other in-vitro models. Finally, they use computational modeling of known AD pathways to narrow down a list of candidate targets, which are selected using mathematical modelling, for testing FDA-approved drugs. They attempt to validate these in some of their models.

While an innovated and much appreciated approach, there are some critical areas of Improvement:

1) Data presented in Figure 3. Localization of the plaques could be clearer, especially the subcellular localization. Perhaps could be complemented by alternate imaging methods that show this key property of the model at higher resolution.

2) Data presented in Figure 4. Potentially move oxidative stress validation to supplemental.

3) Data presented in Figure 6. Park et al. tested FDA-approved drugs on both LPLA288T SNP PiB+ iCOs and E4iso iCOs as a validation of their network-based drug screening platform. Results of the validation experiments were not entirely conclusive due in part to lack of a clear dose-dependent relationship. These results therefore would require further validation, potentially by testing with more replicates and in the other cell lines. Drug testing was not performed on PiB+ iCOs that contain the ApoE ϵ 4 allele. In addition to LPLA288T SNP PiB+ iCOs only representing a small subset of the iCOs used in this study as this SNP is not present in other PiB+ iCOs, these iCOs also do not carry the ApoE ϵ 4 allele. Furthermore, although E4iso iCOs were used for drug testing, they are not derived from AD patient iPSCs. Thus, to properly validate their model for precision medicine applications, FDA-approved drugs need to also be tested on PiB+ iCOs that carry the ApoE ϵ 4 allele. These COs better represent the genetic background of human sporadic AD patients than E4iso iCOs, while also carrying the ApoE ϵ 4 allele missing in the LPLA288T SNP PiB+ iCOs.

While I appreciate the unique aspect the mathematical model brings to the paper, I also felt the flow of the paper would be much improved if this was more concisely summarized in the main text.

Reviewer #4:

Remarks to the Author:

The manuscript submitted describes a high-content drug-screening platform (HCS) that uses human iPSC-derived cerebral organoids (iCOs) in order to identify Alzheimer's disease (AD) drug candidates. For this purpose, pluripotent stem cells (iPSC) derived from participants were used that had been selected on the basis of the presence or absence of pathological features of AD or the ApoE4 genotype. In addition to the use of iCO's from participants with preclinical sporadic AD, a molecular regulatory network model for AD was developed and integrated in order to filter suitable candidates from a library of FDA-approved drugs, the efficacy of which was then tested in the HCS-system.

From a technical point of view the described platform contains a couple of interesting and important new features that might help to overcome some of the present limitations in the use of organoids for high-content drug screening purposes. One of them is a stringent emphasis on quality control with respect to producing a large number of homogenous well shaped and evenly sized iCO's. Demonstrating the feasibility of obtaining such iCOs will be relevant for many other applications of organoids and of interest to others in the community.

The selection and comparison of iCO's from participants with or without a high burden of A β , total tau and phosphorylated tau deposits, is also an important element of the screening platform presented, providing a potentially more relevant disease model for sporadic AD than those available so far.

Furthermore, the integration of a molecular regulatory network model for AD into the screening platform is put forward by the authors as another key innovative element that will facilitate the identification of suitable drug candidates. The approach nicely exploits multiple sources of information: It takes both gene expression data from iPSC models and prior knowledge from curated AD pathways into consideration, and also prefilters the candidate drugs by removing those that do not pass the blood-brain barrier (BBB).

However, it is not clear that the network analysis in this approach provides much additional and robust filtering information for the candidate drug selection: Instead of performing an unbiased genome-scale network analysis, the authors have already pre-selected a network of only 77 nodes with well-known functional AD associations, mainly from existing curated AD pathways in KEGG AlzPathway. This limits the selection of candidate drug targets significantly - the 77 nodes cover already well-studied AD protein targets, and due to this pre-selection, a subset of them will always be chosen as candidate targets, independent of the observed alterations in the iPSC transcriptomics data.

The known drug associations of these 77 nodes, and the drug BBB-filter will further reduce the number of candidate targets and compounds, again independent of the content of the iPSC data. Thus, before applying any network analysis, the drugs are already pre-filtered to those that have previously been described to target the known AD-associated proteins in the AlzPathway map or the KEGG Alzheimer pathway and that pass the BBB. These will all be reasonable candidate drugs, but with limited novelty.

The network analysis is used as an additional filter to further reduce the candidate drug selection, but it is unclear whether it provides significant and robust filtering information beyond the previous generic filters. The two main reasons why there are doubts regarding the robustness and reliability of this additional filtering step are outlined in the comments 1) and 2) below: The small number of RNA-seq experiments limits the statistical power, the transcriptomics data comes from only 11 participants, which may lead to subject-specific idiosyncrasies (1), and the manuscript suggests that the significance scores to determine the differentially expressed genes may not have not been adjusted for multiple hypothesis testing and/or a relaxed p-value cut-off was used.

Main comments:

1.) One limitation is that the data comes from only 11 participants. The data for these subjects may not be representative for the overall population of sporadic AD patients, and subject-specific idiosyncrasies in the data that are not disease-relevant could lead to errors in the transcriptome-based network model and perturbation analysis. The authors mention that they aim at a personalized precision medicine approach, but the distinction between disease-associated biological variance and other sources of variance in the data is still an issue with small numbers of subjects, and small numbers of RNA-seq experiments limiting the statistical power, in particular for the transcriptome-derived network analysis. It is therefore recommended to compare the data against larger public iPSC data for AD, e.g. derived from the GEO database for the transcriptomic analyses.

2.) It is not clear whether the p-values that were used to define the differentially expressed genes (DEGs) were adjusted for multiple hypothesis testing. Using a relaxed p-value cut-off for the pathway analysis ($-\log_{10}(\text{p-value}) > 1$, corresponding to a p-value cut-off of 0.1) can be justified by taking into consideration that pathways with an enrichment of many small, close-to-significant alterations are likely disease-relevant. However, since the authors do not only aim at ranking GO processes, but also at identifying individual drug target genes, a false-discovery-rate (FDR) above 10% for individual genes would lead to too many errors (in particular, if the authors did not use FDR-adjusted p-values, but nominal p-values). Therefore, FDR corrected p-values should be used for all analyses that rely on the significance of individual gene alterations.

3) The authors state that the pre-selected network consists of 77 nodes and 203 links. While these nodes representing mainly genes/proteins from KEGG and AlzPathway definitely play important roles in AD, a significantly larger number of genes/proteins will likely be relevant for AD than this pre-filtered subset, and the restriction to mostly KEGG/AlzPathway-derived nodes may bias the results towards target genes in these pathways that are already well-known and whose associated drugs therefore have limited novelty. A possibility to avoid this limitation is to use a pathway-agnostic network analysis (e.g. using a genome-scale network from the STRING web-service or other public resources for genome-scale gene regulatory or protein interaction networks) to identify network clusters of transcriptomic alterations, which are not already captured by the known pathway definitions.

4) To show that the mathematical modeling / network analysis provides a significant added value beyond the compound filtering obtained from the network model pre-selection of 77 nodes and the BBB-filter, it would be useful to compare the ranked target and compound lists with and without the additional network analysis (e.g. testing whether there is an improved enrichment of known AD protein drug targets, such as BACE1, MAOB, MAPT etc., that have been considered in AD clinical trials, in the network analysis derived ranking list).

In summary, the current manuscript describes the use of a drug screening platform that overcomes some of the limitations of current iPSC/iCO human disease models and is of potential interest beyond AD. However, it does not provide sufficient information to show that network analysis improves the filtering significantly beyond what is already achieved by the prior generic filtering steps. Some possibilities to adjust the analysis workflow (e.g. by using a genome-scale network analysis approach) are suggested.

**Response to the Reviewers' Comments and Summary of Changes**

***To reviewers: The Line numbers are based on the automatically-converted pdf**
**file not the original manuscript word File (.docx).**

**Response to the specific comments of Reviewer 1:**

**[COMMENT #1]**

Research reproducibility of the modeling results: While the Boolean functions of the
model are available as plain text in the Supplementary Excel file, the model should be
shared in the SBML-qual format (PMID: 24321545) to make it easier for others to
reproduce the presented research, and to further expand the model. At a minimum, this
SBML file should be available as part of this manuscript. Better would be making the
model available in any or all of existing Boolean model repositories (e.g., BioModels
(which has been supporting Boolean models for a while now; PMID: 29106614),
GINsim.org, Cell Collective; PMID: 22871178).

**[RESPONSE]**

Following the reviewer's comment, we have converted our Boolean model to SBML-
qual format, and the SBML file is available at [[https://github.com/syjang-](https://github.com/syjang-SBiE/Alz_neuron_network)
[SBiE/Alz_neuron_network](https://github.com/syjang-SBiE/Alz_neuron_network)] (see 'Methods – Mathematical modeling' of the revised
manuscript for details, **line 631-632**). The Boolean functions of our model were written
in accordance with the BoolNet format, and they have been converted to SBML-qual
format using BioLQM v0.6.1. As soon as our paper is accepted, we will add detailed
annotations to our model and upload its SBML file to the existing Boolean model
repositories such as BioModels, GINsim.org, CellCollective by referencing our study.

**[COMMENT #2]**

Related to reproducibility, I don't believe the methods for the modeling component are
comprehensive enough. For example, what initial state(s) did the authors use for the
quality control input-output analyses (**Fig. 5 and Fig. 5-1**), and the actual predictions?
Did the authors perform initial condition sensitivity analysis?

**[RESPONSE]**

For the quality control input-output analyses (**Supplementary Fig. 7-1, 7-2**), the initial
state of all but three nodes were set to zero: the value of 'ROS' node was set according
to input conditions and the values of 'APOE4' and 'LPL' were fixed depending on the
given genetic conditions. For given genetic conditions, qualitative input-output analysis
was performed for all 1,000 input levels of 'ROS' between 0% and 100% over 0.1%
increment. For each input level, the Boolean network simulation was conducted over
1,000-time steps. Each node's state value during the last 300-time steps were tracked,
and the average node state value in this period was taken as the 'average node activity'.

For the actual prediction (**Fig. 5**), we used the ‘getAttractors’ function in the
‘BoolNet’ package. Due to the high computational cost of simulating all initial states,
we randomly sampled one million initial states. We confirmed that the main results
were not sensitive to the randomly selected initial states through repeated sampling
processes. The function’s ‘method’, ‘type’ parameters were set to ‘random’ and
‘synchronous’, respectively to find attractors. We also used ‘genesON’ and ‘genesOFF’
parameters of the function to simulate genetic conditions and perturbations.

Following the reviewer’s comment, we have further performed initial condition
sensitivity analysis of the input-output analyses. From the total of 77 nodes, a certain
number of nodes except ‘ROS’, ‘APOE4’ and ‘LPL’ have been randomly selected and
given an initial state value of ‘1’ and the others have been set to ‘0’. The number of
selected nodes has been set from 10 to 70 with 10 nodes interval, and the input-output
analysis has been performed for these randomly generated initial state values. The
average node activities obtained from this analysis have been compared with the
previous results using the ‘pcc’ function of the ‘sensitivity’ R package. We have set the
value of the ‘nboot’ parameter of the function to 100. The results of this initial condition
sensitivity analysis of our Boolean model have all showed Pearson correlation
coefficients of 1. This is because, when we fixed the value of ‘APOE4’ or ‘LPL’ node
for given genetic conditions, all initial states of each simulation trial have converged to
only one attractor. Consequently, we have confirmed that the results of input-output
analyses are not affected by initial conditions. We have added the aforementioned
description to ‘Input-output relationships of the network model’ in ‘Methods’ of the
revised manuscript (**line 658-670**).

**[COMMENT #3]**

For the oxidative stress input-output analyses: what activity levels were used for other
external variables (e.g., Reelin, APOE4, etc.)? Did the authors do these analyses for
multiple external conditions?

**[RESPONSE]**

In our study, ‘APOE4’ was the only external variable considered in this network model.
‘APOE4’, which stands for Apolipoprotein isoform 4, is a well-known risk factor of
Alzheimer’s disease¹. The activity level of ‘APOE4’ was set to have a value of ‘0’ or ‘1’,
depending on the given genetic condition. If APOE isoform is ε4 allele, the activity
level of ‘APOE4’ was fixed to have a value of ‘1’, otherwise it was fixed to have a
value of ‘0’. For the case of ‘Reelin’, we found that the negative link from ‘Apoptosis’
was accidentally missing in the process of visualizing the Boolean model from the
original logic table. The relationship between ‘Reelin’ and ‘Apoptosis’ was enlisted as
‘Reelin = !Apoptosis’ in the original **Supplementary Table 2** and was used in all
processes of obtaining simulation results. In the revised manuscript, we have updated
the figure for our Boolean model (**Fig. 4**) in accordance with the original logic table
(**Supplementary Table 2**).

**[COMMENT #4]**

It is not clear how oxidative stress ("ROS") in the model was used for input-output
analyses with varying activity levels, because the model (**Fig. 4 and Supplementary**
**Table 2**) shows that several other model components regulate ROS -- in other words, it
is not an independent variable (input) that could be easily used for input-output analyses.

**[RESPONSE]**

In our study, oxidative stress ('ROS') was used as an input to describe the effects of
aging, which is a major risk factor of Alzheimer's disease that increases pathological
protein levels (i.e. beta-amyloid, p-tau, etc.)². However, because biological evidence
that ROS is produced by beta-amyloid or Ca²⁺ ion exists, the regulatory interactions
directed to ROS could not be ruled out in our model³. Thus, as the reviewer pointed out,
ROS is not strictly an independent variable in our model. Despite this issue, we used
ROS as an independent variable to analyze the changes of output node states for fixed
initial ROS levels only for input-output analysis.

**[COMMENT #5]**

The authors did an excellent job annotating the model with citations specific to the
model components' interactions. This level of annotations will aid in the transparency
of the model. For readability issues, can the authors replace the PMIDs in all
supplementary documents with actual citations and references?

**[RESPONSE]**

Following the reviewer's comment, we have revised the manuscript by replacing the
PMIDs in all supplementary documents with actual citations and references.

**[COMMENT #6]**

In addition to providing more details about the modeling methodology and
model/simulation set-up, it would be helpful if the authors included citations to offer
readers with options further to dive even deeper into Boolean/logical modeling.
Examples of more recent reviews on the topic include PMID: 27303434, 32313939, etc.
BoolNet should be cited. Finally, the input-output analysis of the model leverages the
conversion of the binary input/output of Boolean networks into semi-continuous
Activity Levels and "%ON" concepts originally developed in Helikar et al., 2008. As
such, it might be appropriate to cite the work.

**[RESPONSE]**

Following the reviewer's comment, we have cited the latest review papers on
Boolean/logical modeling in the revised manuscript (**line 618-619**). A citation for the
BoolNet package has also been added (**line 627-631**). Finally, a citation for the Helikar
et al., 2008 has also been added to the 'Methods' section of the revised manuscript, so
that appropriate citation for the previous study is made (**line 648**).

[COMMENT #7]

Some of the figures are very complex, making them hard to read without zooming in
significantly (Fig. 6 is the least readable)

[RESPONSE]

We have increased the font size on our figures and especially modified Fig. 6. The
graph type has been changed in order that the medicinal effects can be seen well at a
glance (all of the cell viability data have moved to Supplementary Fig. 11. Instead,
PiB⁺ #5 iCOs data were added in the Fig. 6 according to the Reviewer 4's comment).

[COMMENT #8]

For the drug effect studies, can the authors please more clearly explain what was/were
the control(s)?

[RESPONSE]

No drug treatment groups (we call them as 'vehicle group'; Veh) were used as control
groups. Since our drug powders were initially diluted in DMSO or distilled water (DW)
to make them dissolve, iCOs of Veh groups receive a treatment with appropriate amount
of DMSO or distilled water (DW) (the same volume of solution used to make drug
dissolve) that do not contain drugs meant to affect iCOs. The detailed methods for the
drug treatment were added to the 'Drug treatment' section which is described in the
Methods section (line 521-526).

[COMMENT #9]

How many replicates were used?

[RESPONSE]

Six replicates (individual iCOs) per each drug dose were performed. We also mentioned
it in the 'Drug treatment' section in the Methods section (line 526-527).

[COMMENT #10]

How did the authors select the drug concentrations?

[RESPONSE]

Since our drug candidates were subjected to drug repositioning, they had mainly been
tested for applications in fields other than neurodegenerative disorders. Most of the
references used cell types different from neurons. We could hardly find any studies
treating the drugs on brain organoids as it is a relatively novel and unconventional
method in drug screening. Consequently, we referred to studies using the drugs in *in*
*vitro* application and optimized for our system by testing the dosage range that includes
the optimal level suggested by the references.

When setting the dosage range, we had two points into consideration. Firstly, a lower
dosage limit should be tested to observe the potential neurotoxic effects of the drugs.
Responsiveness to drugs might differ according to cell types, and iPSC-derived neurons,
which resemble primary cells, could be more vulnerable to a higher concentration than

conventional cell lines. Secondly, the fact that brain organoids are three-dimensional
aggregates of interconnected neurons with complex cytoarchitecture and extracellular
matrix should be considered. These might serve as barriers to simple diffusion of drugs
and induce essentially different drug penetration dynamics and subsequent uptake by
cells compared to conventional monolayer culture. It led us to test the upper
concentration limit as equal to or higher than the references.

For example, Ripasudil, which is approved for glaucoma and ocular hypertension
treatment, has been treated at a concentration of 10 μM on human corneal endothelial
cells on the previous studies^{4,5}; we tested various conditions ranging from 0.25 μM to
10 μM to account for potential neurotoxicity. Everolimus, an mTOR inhibitor, has been
treated on human coronary artery endothelial cells at 0.5 μM ⁶, and on A-498 and Caki-1
cell lines at 1 μM ⁷. Thus, we set a dosage range spanning from 0.25 μM to 5 μM that
can include the optimal dosage level indicated by the references. After setting the
concentration range to be tested, we further validated it by MTT assay to observe its
influence on cell viability. Although the drug effects were trivial on enhancing cell
viability, we could not observe any harmful effects induced by the drugs at our treated
doses.

[COMMENT #11]

Why did the authors use 24h for the drug treatment? Did the analysis post-treatment
happen at the 24-hour mark, or was it later? Why not use 48 or 72 hrs?

[RESPONSE]

Although the ‘Veh’ groups (used as controls whereas the treatment groups were used as
cases) were not treated with the drugs, they were simultaneously grown in the opti-
MEM media, which is not suitable for the permanent cell culture of iCOs, during the
treatment of drugs with other iCOs. We thought the best time point will be that of
‘showing a natural tendency of cell death but not too much’. We speculated 48 or 72h
(>24h) may be too long for the drug test because the ‘Veh’ groups also can show
excessive loss of neuronal cells (might be called as ‘cell senescence’) even without the
drugs. As expected, the Veh group showed statistically significant loss of their own cell
viability from 48h timepoint (reference figure below). So, we decided to use 24h
timepoint for the drug treatment.

<Reference figure for COMMENT #11>

**[COMMENT #12: Minor concerns]**
* Line 45: remove "the" before precision medicine/ * Line 50: remove "the" before
symptoms/ * Line 76: "in vitro" should be italicized/ * Line 99: "Mathematic" should be
"Mathematical"/ * Line 128: Missing "genes" from "differentially expressed genes
(DEG) patterns"/ * Line 197: Remove "comprehensive" -- while the model is not a "toy
model," comprehensive is a relative term, and I don't believe this model is
comprehensive (for example many other pathways known to interact with the modeled
pathways could be included). Perhaps "relevant" is a more appropriate word/ * Line 313:
Similar to above, I recommend removing "the whole" because the model does not
include the entire interaction network /* Line 321: "was focusing" does not seem
grammatically correct; perhaps change to "was to focus on."?/ * Line 333: remove "but"

**[RESPONSE]**

We corrected all of them. Thanks for the kind and detailed comments.

**Response to the specific comments of Reviewer 2:**

**[COMMENTS]**

This manuscript entitled “A network-based drug-screening platform for Alzheimer’s
disease by integrating mathematical modeling and pathological features of human brain
organoids” describes a network-based drug-screening platform developed by integrating
mathematical modeling and the pathological features of Alzheimer’s disease (AD) with
human iPSC-derived cerebral organoids. Basic idea of this manuscript is a novel and
interesting in developing drug-screening for Alzheimer’s disease.

However, authors should clearly describe their methods in construction of AD
molecular network. ^[1]How did they narrow genes/proteins/phenotypes down to only 77
genes/proteins/phenotypes? It appears to be an arbitrary choice. At least, AlzPathway
consists of 1,347 species (genes and proteins) and 129 phenotypes. ^[2]Authors need to
explain the reason why they focus on MAPK signaling pathway, WNT signaling
pathway, and PI3K-AKT signaling pathway as molecular regulatory network of AD
(**Fig. 4A**) which is a basis for network analysis in this manuscript. ^[3]By the way, **Fig.**

**4A** is the correct figure? For example, Reelin has relationships with not only Apoer2 but
also VLDLR and Apoptosis according to a logic table (**Supplementary Table 2**).

232 ^[4]Authors also should clearly describe their methods in analysis of the AD network
model and identification of candidate drugs. For example, **Fig. 5A** illustrates up-
regulated and down-regulated pathways according to their perturbation analysis, but
they did not explain the definition of “up-regulated” and “down-regulated”
pathways, and ^[5]they did not show their results of perturbation analysis. ^[6]Authors also
should clearly describe their methods of attractor landscape analysis and unfortunately

238 ^[7]attractor landscape drawn in **Figure 5B** is too small to see. ^[8]By the way, the idea of
239 phenotype score is interesting to estimate the pathological level, but there are several
concerns. Authors choose key proteins and phenotypes such as A β , p-tau, synapse loss,
apoptosis, and autophagy for calculation of phenotype score, but how to choose these
key proteins and phenotypes? ^[9]Phenotype score looks work well but p-tau proteins
have double impact on the score because p-tau activates synapse loss. They developed a
network model but they did not consider these kinds of network effects in the
calculation of phenotype score. If phenotype score well works to estimate the
pathological level, ^[10]why did they need to develop a network model and conduct a
perturbation analysis? Why don’t authors directly conduct the phenotype score
calculation?

Basic idea of this manuscript is a novel and interesting in developing drug-screening for
Alzheimer’s disease, but they need to clearly describe their methods and results.

**[COMMENT #1]**

How did they narrow genes/proteins/phenotypes down to only 77
genes/proteins/phenotypes? It appears to be an arbitrary choice. At least, AlzPathway
consists of 1,347 species (genes and proteins) and 129 phenotypes.

**[RESPONSE]**

The AlzPathway has various types of components (e.g. gene, protein, signaling pathway,
*etc.*), so its components could not be directly used as nodes for the network model. For
this reason, we used the AlzPathway for sorting out the list of signaling pathways. Of
the components in the AlzPathway, only the signaling pathways of the neuron
(indicated by 'n_' prefix) were considered for the construction of the network model.
These pathways were grouped into Notch, RELN, MAPK, Jak/Stat, Wnt, NR1/NR2R,
Ca²⁺, TGFβ and mTOR (PI3K-AKT) signaling pathways. Among them, TGFβ signaling
pathway related to microglia⁸ was excluded. Based on these signaling pathway
information, we searched for experimental studies in neuronal context to the best of our
knowledge. By using this literature-based approach, we reconstructed simple regulatory
links (i.e. activate or inactivate) and actual regulatory relationships such as update logic
of node state (e.g. A node = B node AND C node) (**Supplementary Table 2**).

**[COMMENT #2]**

Authors need to explain the reason why they focus on MAPK signaling pathway, WNT
signaling pathway, and PI3K-AKT signaling pathway as molecular regulatory network
of AD (**Fig. 4a**) which is a basis for network analysis in this manuscript.

**[RESPONSE]**

In our study, we focused on APOE4, LPL-related signaling pathways in line with the
experimental results in the main manuscript (**Fig. 2**). Through a literature-based
approach, we constructed the model network primarily based on MAPK, Wnt, and
mTOR signaling pathways⁹⁻¹¹, and also considered other signaling pathways in the
AlzPathway. We have added more explanation on the model construction in the revised
manuscript (**line 608-612**).

**[COMMENT #3]**

By the way, **Fig. 4a** is the correct figure? For example, Reelin has relationships with not
only Apoer2 but also VLDLR and Apoptosis according to a logic table
(**Supplementary Table 2**).

**[RESPONSE]**

As the reviewer pointed out, there was a typographical error in **Fig. 4a** of the original
manuscript. In the revised manuscript, we have corrected this and updated the figure for
our Boolean model in accordance with the original logic table.

**[COMMENT #4 & 6]**

Authors also should clearly describe their methods in analysis of the AD network model
and identification of candidate drugs. For example, **Fig. 5a** illustrates up-regulated and

down-regulated pathways according to their perturbation analysis, but they did not
explain the definition of “up-regulated” and “down-regulated” pathways.

Authors also should clearly describe their methods of attractor landscape analysis

**[RESPONSE]**

As we described in the ‘Mathematical modeling’ section of ‘Methods’, attractor is a
network state that represents a specific biological phenotype. Attractor landscape is the
landscape of all attractors and the set of all states converging to each attractor (basin of
attraction) in the state space of a given network model. Attractor landscape analysis
enables quantitative evaluation of the network system by transforming complex
dynamical properties of the network model into the states of convergence (attractors)
and the converging propensity of initial states (basins). Using attractor landscape
information, we can quantify node activities by averaging attractor states weighted by
their basin sizes. In other words, attractor landscape analysis is a kind of procedure that
can quantify node activities using attractor landscape information and further estimate
the perturbation effect *in silico*. We have added the aforementioned description to
‘Attractor landscape analysis’ in ‘Methods’ of the revised manuscript (**line 633-641**).

The term ‘up-regulated’ and ‘down-regulated’ means the increase and decrease,
respectively, of representative nodes (genes or proteins) for each signaling pathway. For
instance, ‘up-regulated’ and ‘down-regulated’ correspond to the increase and decrease,
respectively, of ERK node activity in case of the MAPK signaling pathway. The
representative nodes for each signaling pathway are described on the right-side table of
the **Supplementary Fig. 8**.

**[COMMENT #5]**

They did not show their results of perturbation analysis.

**[RESPONSE]**

The major perturbation analysis results used in the experiments (**Fig. 5**) were provided
as a bar graph in the **Supplementary Fig. 10**. Other results could not be included in the
Supplementary Dataset because the list of candidate targets is too long (e.g. the number
of double perturbations: ${}_{70}C_2=2,416$ cases). Instead, we provided the R-code for the
perturbation analysis in the ‘Methods’ of the revised manuscript (**line 631-632**).

**[COMMENT #7]**

attractor landscape drawn in **Figure 5b** is too small to see.

**[RESPONSE]**

Following the reviewer’s comment, we have redrawn the ‘Attractor landscape’ in Fig. 5
of the revised manuscript.

In the conceptual image of the ‘① Node perturbation’ box of **Fig. 5a**, each grey circle
in ‘Attractor landscape’ represents network state, which is a collection of node states (1
or 0). The black circle is the attractor to which all states within the basin of attraction

converge. The grey circles within the boundary of attractor A belong to the basin of A.
The basin size of attractor A, which is the number of grey circles in the basin of A, is
larger than the basin size of attractor B.

**[COMMENT #8]**

By the way, the idea of phenotype score is interesting to estimate the pathological level,
but there are several concerns. Authors choose key proteins and phenotypes such as A β ,
p-tau, synapse loss, apoptosis, and autophagy for calculation of phenotype score, but
how to choose these key proteins and phenotypes?

**[RESPONSE]**

The reasons why we chose A β , p-tau, neuron loss, synapse loss, and autophagy as key
proteins and phenotypes are as follows. First, because A β and p-tau are the well-known
biological markers of Alzheimer's disease, they were chosen as the target proteins the
expression levels of which should be decreased. Second, the goal of our study is to
identify an optimal target that can decrease the neuron/synapse loss, which is associated
with the cognitive impairment in Alzheimer's disease, by analyzing the network model.
Third, based on previous findings that the degradation of autophagy function increases
accumulation of A β , we included autophagy as a key phenotype^{12,13}. Choosing
autophagy as a key phenotype helps to understand whether the accumulation
mechanism of A β is due to an increase of A β production or a decrease in autophagy
function. For these reasons, we chose A β , p-tau, neuron loss, synapse loss, and
autophagy as key proteins and phenotypes in this study.

As we described in the 'Analysis of the AD network model and identification of
candidate drugs' part of 'Results', we calculated the phenotype score using the key
proteins and phenotypes' node activities obtained from attractor landscape analysis of
our network model. Therefore, simulation of the network model is a prerequisite to
obtain the phenotype score. In the network model with a given genetic condition (e.g.
APOE4, LPL), we can identify which node contributes to producing a desirable
phenotype score using the node perturbation analysis as described in 'Result' of the
main manuscript. The desirable phenotype score refers to the case where the activities
of the pathological proteins and phenotypes are low. Hence, both perturbation analysis
and calculation of the phenotype score are required to find the optimal target(s) for a
given genetic condition.

**[COMMENT #9]**

Phenotype score looks work well but p-tau proteins have double impact on the score
because p-tau activates synapse loss. They developed a network model but they did not
consider these kinds of network effects in the calculation of phenotype score.

**[RESPONSE]**

Although there is a positive link from 'p-tau' to 'Synapse loss', activation of 'p-tau'
alone cannot satisfy the sufficient condition for activating 'Synapse loss' according to

its update logic; (synapse_loss = (Cofilin | tau_p) & !(BDNF | CREB)). For this reason,
there is no double impact of 'p-tau' on the phenotype score.

**Response to the specific comments of Reviewer 3:**

**[COMMENT #1]**

Data presented in **Figure 3**. Localization of the plaques could be clearer, especially the
subcellular localization. Perhaps could be complemented by alternate imaging methods
that show this key property of the model at higher resolution

**[RESPONSE]**

As the reviewer suggested, we have tried imaging brain organoid sections in higher
resolution (40X) with Zeiss LSM 700 (**Supplementary Fig. 5, line 192-196**). The
images clearly show that amyloid-beta aggregates are formed in extracellular interstitial
spaces, and hyper-phosphorylated tau co-localizes intracellularly along with neuronal
marker MAP2. As sites of amyloid deposition and tau hyper-phosphorylation do not
always overlap, it is indefinite to conclude by a single image that levels of pathogenic
protein differ significantly between two groups. Thus, we used a total of 64 organoid
sections from 6 individual lines (CN1, CN5, AD4, AD5, E3^{par}, E4^{iso}) that came from
various Z-positions of an organoid to reflect diverse regions of a three-dimensional (3D)
object. In accordance with the main results, two-dimensional (2D)
immunohistochemistry images show that amyloid-beta and p-tau immunoreactivity was
significantly increased in PiB positive group compared to the negative group. Also,
isogenic E4 organoids retained higher levels of amyloid-beta and p-tau compared to the
parental E3 line. Even though we have used many samples for quantification, 2D
segmentalization of a 3D object could introduce unintentional bias where some regions
are over-represented. This was the foremost reason we used 3D tissue clearing
technology and ImageXpress Micro Confocal system to image organoids and measure
pathogenic protein levels as it provides a holistic view on an entire organoid.

**[COMMENT #2]**

Data presented in **Figure 4**. Potentially move oxidative stress validation to
supplemental.

**[RESPONSE]**

The data presented in **Fig. 4** show the molecular regulatory network model of
Alzheimer's disease and its validation for different oxidative stress levels. Following the
reviewer's comment, we have repositioned previous **Fig. 4c to Supplementary Fig. 9**
(the result of enriched pathway analysis) to Supplementary data of the revised
manuscript. Since validation of network model is as important as building the model in
Boolean/logical modeling studies¹⁴, we have kept **Fig. 4b** to show essential input-output
relationships of the network model in the revised manuscript.

**[COMMENT #3]**

Data presented in **Figure 6**. Park et al. tested FDA-approved drugs on both LPLA288T
SNP PiB+ iCOs and E4iso iCOs as a validation of their network-based drug screening

platform. Results of the validation experiments were not entirely conclusive due in part
to lack of a clear dose-dependent relationship. These results therefore would require
further validation, potentially by testing with more replicates and in the other cell lines.
Drug testing was not performed on PiB⁺ iCOs that contain the ApoE ε4 allele. In
addition to LPLA288T SNP PiB⁺ iCOs only representing a small subset of the iCOs
used in this study as this SNP is not present in other PiB⁺ iCOs, these iCOs also do not
carry the ApoE ε4 allele. Furthermore, although E4^{iso} iCOs were used for drug testing,
they are not derived from AD patient iPSCs. Thus, to properly validate their model for
precision medicine applications, FDA-approved drugs need to also be tested on PiB⁺
iCOs that carry the ApoE ε4 allele. These COs better represent the genetic background
of human sporadic AD patients than E4^{iso} iCOs, while also carrying the ApoE ε4 allele
missing in the LPLA288T SNP PiB⁺ iCOs.

**[RESPONSE]**

We totally agree with that. As the reviewer suggested, we further performed drug test
by using the iCOs from PiB⁺ carrying the ApoE ε4 allele (PiB⁺ #5 iCOs). We added this
result in **Fig. 6d** (We also changed graph types in **Fig. 6** in order that the medicinal
effects can be seen well at a glance). Some interpretations are as follows:

i) Most of the drugs showed better efficacy in reducing A β and tau deposition as their
concentration increased, except for the ‘Astaxanthin’ and ‘Everolimus with 1 uM
Flibanserin’. ii) PiB⁺ #5 iCOs showed better drug effectiveness on the A β reduction
than E4^{iso} iCOs. One possibility is that it is because PiB⁺ #5 iCOs are basically derived
from sporadic AD participants, but E4^{iso} iCOs are not although we cannot rule out the
possibility of difference on genetic background from each patient. iii) Similar to E4^{iso}
iCOs, the combinative therapy using Flibanserin with Ripasudil worked best when
compared to other drugs.

Although, the tendency from PiB⁺ #5 iCOs is not exactly matched with E4^{iso} iCOs, it is
obvious that most of the drugs had effectiveness on the reduction of A β and tau.
Furthermore, we think it is quite natural because PiB⁺ #5 iCOs are basically derived
from sporadic AD participants, but E4^{iso} iCOs are not, as mentioned above. Therefore,
these results indicate that we validated our network-based drug-screening platform by
integrating mathematical modeling and pathological traits of human iCOs.

**[COMMENT #4]**

While I appreciate the unique aspect the mathematical model brings to the paper, I also
felt the flow of the paper would be much improved if this was more concisely
summarized in the main text.

**[RESPONSE]**

We have modified our discussion section (**line 330-340**) and results section (**line 200-**
**202, line 212-220**) to make it clearer as you recommended.

**Response to the specific comments of Reviewer 4:**

**[COMMENT #1]**

One limitation is that the data comes from only 11 participants. The data for these
subjects may not be representative for the overall population of sporadic AD patients,
and subject-specific idiosyncrasies in the data that are not disease-relevant could lead to
errors in the transcriptome-based network model and perturbation analysis. The authors
mention that they aim at a personalized precision medicine approach, but the distinction
between disease-associated biological variance and other sources of variance in the data
is still an issue with small numbers of subjects, and small numbers of RNA-seq
experiments limiting the statistical power, in particular for the transcriptome-derived
network analysis. It is therefore recommended to compare the data against larger public
iPSC data for AD, e.g. derived from the GEO database for the transcriptomic analyses.

**[RESPONSE]**

As reviewer suggested, we have tried to compare our transcriptome data to the public
GEO database (**Supplementary Fig. 4, line 164-175, line 595-599**). It is not possible to
find public iPSC- derived brain organoid data for AD, but we found a public iPSC-
derived neuron data (PIN) for AD (Accession number: GSE143951, Platform number:
GPL16043) and public human AD brain data (PHB) (Accession number: GSE109887,
Platform number: GPL10904) (**Supplementary Fig. 4a-4b**). We compared PHB GO
data from Toppgene DB (<https://toppgene.cchmc.org>) to PIN or our own transcriptome
data, to reveal that our transcriptome data from human iCOs is more similar to PHB
than PIN. We performed GO similarity analysis to identify how many GO terms
overlapped each other. We used three GO sub-ontologies (CC, cellular components; BP,
biological processes; MF, molecular functions) (cut-off, FDR- corrected p-value < 0.05).
Interestingly, our own transcriptome data had dramatically high GO similarity to PHB
(CC, 64.9% for PiB iCOs and 76.5% for E4 iCOs; BP, 58.7% for PiB iCOs and 26.0%
for E4 iCOs; MF, 46.7% for PiB iCOs and 29.4% for E4 iCOs), whereas PIN had low
GO similarity to PHB (CC, 7.0% for PIN; BP, 4.2% for PIN; MF, 8.3% for PIN)
(**Supplementary Fig. 4c**). Specifically, we have tried to confirm that our significant GO
terms (**Fig. 2i**) are also included in PHB or PIN. Most of the GO terms were also
included as significant terms in PHB (21/28, 75%), but not in PIN (6/28, 21%)
(**Supplementary Fig. 4d**). Therefore, we concluded that our own transcriptome data is
closer to the real human brain transcriptome data.

**[COMMENT #2]**

It is not clear whether the p-values that were used to define the differentially expressed
genes (DEGs) were adjusted for multiple hypothesis testing. Using a relaxed p-value
cut-off for the pathway analysis ($-\log_{10}(\text{p-value}) > 1$, corresponding to a p-value cut-off
of 0.1) can be justified by taking into consideration that pathways with an enrichment of
many small, close-to-significant alterations are likely disease-relevant. However, since

the authors do not only aim at ranking GO processes, but also at identifying individual
drug target genes, a false-discovery-rate (FDR) above 10% for individual genes would
lead to too many errors (in particular, if the authors did not use FDR-adjusted p-values,
but nominal p-values). Therefore, FDR corrected p-values should be used for all
analyses that rely on the significance of individual gene alterations.

**[RESPONSE]**

Thank you for the critical comment. It needs to be clarified that different correction
strategies were adopted for each RNA-seq analysis.

First, we already applied the FDR corrected p-values for the validation of mathematical
model and identification of individual drug target genes (**Fig. 5b and Supplementary**
**Fig. 8,9**). We have mentioned this in the legend of **Supplementary Fig. 8**.

Second, to demonstrate that gene expression profiles (GO analysis) differ between PiB⁻
and PiB⁺ groups (**Fig 2i and Supplementary Fig 2a, c, f**), we subjected five samples
from each group, performed RNA-seq on each sample and compared gene expression
profiles (GO analysis) by using the selected DEGs. For this specific analysis, we did not
apply multiple correction for the selection of DEGs because we intended to be more
inclusive in documenting subtle differences in gene expression changes, as the reviewer
pointed out. However, since we had not applied FDR correction, even for the GO
analysis deriving significant GO terms from the selected DEGs, we have changed p-
values to FDR- adjusted p-values (**Fig 2i and Supplementary Fig. 3**). We have
mentioned this in the legend of **Fig 2 and Supplementary Fig. 3**

Regarding the reviewer's comment, we have also clearly mentioned it in **Methods**
section (**line 592-595, line 600-603**), please refer to the **Methods** section.

**[COMMENT #3]**

The authors state that the pre-selected network consists of 77 nodes and 203 links.
While these nodes representing mainly genes/proteins from KEGG and AlzPathway
definitely play important roles in AD, a significantly larger number of genes/proteins
will likely be relevant for AD than this pre-filtered subset, and the restriction to mostly
KEGG/AlzPathway-derived nodes may bias the results towards target genes in these
pathways that are already well-known and whose associated drugs therefore have
limited novelty. A possibility to avoid this limitation is to use a pathway-agnostic
network analysis (e.g. using a genome-scale network from the STRING web-service or
other public resources for genome-scale gene regulatory or protein interaction networks)
to identify network clusters of transcriptomic alterations, which are not already captured
by the known pathway definitions.

**[RESPONSE]**

The AlzPathway has various types of components (e.g. gene, protein, signaling pathway,
*etc.*), so its components could not be directly used as nodes for the network model. For
this reason, we used the AlzPathway for sorting out the relevant signaling pathway lists.
Of the components in the AlzPathway, only the signaling pathways of the neuron

(indicated by ‘n_’ prefix) were considered for the construction of the network model.
These pathways were grouped into Notch, RELN, MAPK, Jak/Stat, Wnt, NR1/NR2R,
Ca²⁺, TGFβ and mTOR (PI3K-AKT) signaling pathways. Among them, TGFβ signaling
pathway related to microglia was excluded.

Based on these signaling pathway information, we searched for experimental studies
only in neuronal context to the best of our knowledge. By using this literature-based
approach, we reconstructed simple regulatory links (i.e. activate or inactivate) and
actual regulatory relationships such as update logic of node state (e.g. A node = B node
AND C node) (**Supplementary Table 2**).

In our study, we focused on APOE4, LPL-related signaling pathways in line with the
experimental results in the main manuscript (**Fig. 2**). Through a literature-based
approach, we constructed the model network primarily based on MAPK, Wnt, and
mTOR signaling pathways⁹⁻¹¹, and also considered other signaling pathways in the
AlzPathway. We have added more explanation to the model construction in the revised
manuscript (**line 608-612**).

The purposes of our study are as follows: 1) to understand complex dynamical
properties of the regulatory network, which cannot be done by simple correlation
analysis among the network components, and 2) to identify optimal candidate targets for
each genetic condition based on quantitative dynamical analysis of the regulatory
network model. As the reviewer pointed out, pathway-agnostic network analysis may
have benefits in finding novel targets. However, since we aim to find the optimal targets
for a given genetic condition rather than to find novel targets, the proposed approach
does not fit our research objectives. Also, public resources such as STRING contain
collective information from various cell-types, so we considered that such kinds of
resources might not be suitable for constructing our neuron-specific network model.
Nevertheless, the reviewer’s comment will be very helpful in our future study to find
out novel targets by further extending our network model to include diverse genetic risk
factors and components. We appreciate the reviewer’s valuable comment in this respect.

**[COMMENT #4]**

To show that the mathematical modeling / network analysis provides a significant added
value beyond the compound filtering obtained from the network model pre-selection of
77 nodes and the BBB-filter, it would be useful to compare the ranked target and
compound lists with and without the additional network analysis (e.g. testing whether
there is an improved enrichment of known AD protein drug targets, such as BACE1,
MAOB, MAPT etc., that have been considered in AD clinical trials, in the network
analysis derived ranking list).

**[RESPONSE]**

There were several network analysis studies that constructed gene or protein interaction
networks using databases such as the STRING and found network clusters/modules of
transcriptomic alterations that are significantly related to diseases. However, the data

used in these studies contained information of various cell types, such as neuron,
microglia, astrocyte, *etc.* So, we considered that such kinds of resources might not be
suitable for constructing our neuron-specific network model. In addition, the results
obtained by pathway-agnostic analysis mostly suggested microglia-related components
as targets¹⁵, so the target list that can be obtained from the pathway-agnostic analysis
would be inappropriate for our research objectives.

We also curated drug targets under clinical trial from the research conducted by
Cummings J. et al.,¹⁶ and took only those included in our model; α -secretase, NMDAR,
MAPT, p38 MAPK- α , APP, PP2B, PPAR- γ . The activities of these nodes were
changed in the direction of beneficial propensity when we perturbed the high phenotype
score targets in our network model. Furthermore, our approach has the advantage of
logical modeling, which allows us to understand the underlying mechanism of the
disease, at the molecular regulatory network level, that occurs differently depending on
genetic conditions (even with the same pathological phenotype), and thus to explore
optimal drug target candidates for each genetic condition.

[References]

- 1. Lin, Y.T., *et al.* APOE4 Causes Widespread Molecular and Cellular
Alterations Associated with Alzheimer's Disease Phenotypes in Human
iPSC-Derived Brain Cell Types. *Neuron* **98**, 1141-1154 e1147 (2018).
- 2. Karch, C.M. & Goate, A.M. Alzheimer's disease risk genes and
mechanisms of disease pathogenesis. *Biol Psychiatry* **77**, 43-51 (2015).
- 3. Chen, Z. & Zhong, C. Oxidative stress in Alzheimer's disease. *Neurosci*
*Bull* **30**, 271-281 (2014).
- 4. Goldstein, A.S., *et al.* Assessing the Effects of Ripasudil, a Novel Rho
Kinase Inhibitor, on Human Corneal Endothelial Cell Health. *J Ocul*
*Pharmacol Ther* (2018).
- 5. Okumura, N., *et al.* Effect of the Rho-Associated Kinase Inhibitor Eye
Drop (Ripasudil) on Corneal Endothelial Wound Healing. *Invest*
*Ophthalmol Vis Sci* **57**, 1284-1292 (2016).
- 6. Fejes, Z., *et al.* Endothelial cell activation is attenuated by everolimus
via transcriptional and post-transcriptional regulatory mechanisms
after drug-eluting coronary stenting. *PLoS One* **13**, e0197890 (2018).
- 7. Nayman, A.H., *et al.* Dual-Inhibition of mTOR and Bcl-2 Enhances the
Anti-tumor Effect of Everolimus against Renal Cell Carcinoma In Vitro
and In Vivo. *J Cancer* **10**, 1466-1478 (2019).
- 8. Yuan, C., *et al.* The age-related microglial transformation in Alzheimer's
disease pathogenesis. *Neurobiol Aging* **92**, 82-91 (2020).
- 9. Huang, Y.A., Zhou, B., Wernig, M. & Sudhof, T.C. ApoE2, ApoE3, and
ApoE4 Differentially Stimulate APP Transcription and Abeta Secretion.
*Cell* **168**, 427-441 e421 (2017).
- 10. Lane-Donovan, C. & Herz, J. ApoE, ApoE Receptors, and the Synapse
in Alzheimer's Disease. *Trends Endocrinol Metab* **28**, 273-284 (2017).
- 11. Menzies, F.M., *et al.* Autophagy and Neurodegeneration: Pathogenic
Mechanisms and Therapeutic Opportunities. *Neuron* **93**, 1015-1034
(2017).
- 12. Mawuenyega, K.G., *et al.* Decreased clearance of CNS beta-amyloid in

- Alzheimer's disease. *Science* **330**, 1774 (2010).
- 13. Lipinski, M.M., *et al.* Genome-wide analysis reveals mechanisms
modulating autophagy in normal brain aging and in Alzheimer's
disease. *Proc Natl Acad Sci U S A* **107**, 14164-14169 (2010).
- 14. Abou-Jaoude, W., *et al.* Logical Modeling and Dynamical Analysis of
Cellular Networks. *Front Genet* **7**, 94 (2016).
- 15. Mostafavi, S., *et al.* A molecular network of the aging human brain
provides insights into the pathology and cognitive decline of
Alzheimer's disease. *Nat Neurosci* **21**, 811-819 (2018).
- 16. Cummings, J., Lee, G., Ritter, A., Sabbagh, M. & Zhong, K. Alzheimer's
disease drug development pipeline: 2020. *Alzheimers Dement (N Y)* **6**,
e12050 (2020).

Reviewers' Comments:

Reviewer #1:

Remarks to the Author:

I think this is a strong manuscript and worthy publishing in Nature Communications. The authors did an excellent job addressing my comments, as well as the suggestions of the other reviewers.

Reviewer #2:

None

Reviewer #3:

Remarks to the Author:

The authors' revised manuscript has incorporated significant changes, including the addition of new data, that sufficiently address our initial concerns and recommended improvements. In addition to imaging iCO sections at higher resolution to show clearer subcellular localization of amyloid beta plaques and phosphorylated tau, the authors also utilize various Z-positions to compare pathogenic protein levels between different iCO groups. Furthermore, the authors have also included suggestions to test drug compounds on PiB+ iCOs that contain the ApoE allele, as this better reflects the genetic background of sporadic AD patients.

The combination of this new data as well as modifying the graph type in figure 6 to show a clearer dose-dependent relationship strengthens the validation of their network-based drug-screening platform for AD.

Reviewer #4:

Remarks to the Author:

Thanks for considering some of the suggestions in the revision of the manuscript. Although my principle concerns related to limitations of the study with respect to statistical power and the constraints of using a limited subset of 77 pre-selected nodes still remain, I do think that the study is worth publishing. I still have the concern that if you go in with a highly selected sub-network and use it as a constraint, you come out with output nodes that have already been validated to a certain extent and the novelty is limited.

Nevertheless, the combination of using a network-based filtering approach and the use of larger scale cerebral organoids (e.g. with and without the ApoE4-allele or ROS challenges) apparently is able to at least support the selection of potentially useful existing drugs. The system might also provide hints for prioritizing drug candidates for further mechanistic and clinical evaluation.

The main impact of the study might be the motivation for the field to combine network based-based and other computational approaches with the iPSC-derived organoid technology, allowing both, in-silico as well as experimental genetic and environmental perturbation studies. In that sense the study will probably have a positive impact on a number of organoid applications.

REVIEWERS' COMMENTS

Reviewer #1 (Remarks to the Author):

I think this is a strong manuscript and worthy publishing in Nature Communications. The authors did an excellent job addressing my comments, as well as the suggestions of the other reviewers.

Reviewer #3 (Remarks to the Author):

The authors' revised manuscript has incorporated significant changes, including the addition of new data, that sufficiently address our initial concerns and recommended improvements. In addition to imaging iCO sections at higher resolution to show clearer subcellular localization of amyloid beta plaques and phosphorylated tau, the authors also utilize various Z-positions to compare pathogenic protein levels between different iCO groups. Furthermore, the authors have also included suggestions to test drug compounds on PiB⁺ iCOs that contain the ApoE allele, as this better reflects the genetic background of sporadic AD patients.

The combination of this new data as well as modifying the graph type in figure 6 to show a clearer dose-dependent relationship strengthens the validation of their network-based drug-screening platform for AD.

Reviewer #4 (Remarks to the Author):

Thanks for considering some of the suggestions in the revision of the manuscript. Although my principle concerns related to limitations of the study with respect to statistical power and the constraints of using a limited subset of 77 pre-selected nodes still remain, I do think that the study is worth publishing. I still have the concern that if you go in with a highly selected sub-network and use it as a constraint, you come out with output nodes that have already been validated to a certain extent and the novelty is limited.

Nevertheless, the combination of using a network-based filtering approach and the use of larger scale cerebral organoids (e.g. with and without the ApoE4-allele or ROS challenges) apparently is able to at least support the selection of potentially useful existing drugs. The system might also provide hints for prioritizing drug candidates for further mechanistic and clinical evaluation.

The main impact of the study might be the motivation for the field to combine network based-based and other computational approaches with the iPSC-derived organoid technology, allowing both, in-silico as well as experimental genetic and environmental perturbation studies. In that sense the study will probably have a positive impact on a number of organoid applications.

→ Thank you very much for the Reviewers' kind words and great additional comments. We also agree with the comment from Reviewer #4, therefore, we will proceed to build an advanced model of data-driven using more experimental data in the future. Thank you very much again for your nice comments so far.